# Evaluation of Conditioning Factors of Slope Instability and Continuous Change Maps in the Generation of Landslide Inventory Maps Using Machine Learning (ML) Algorithms

Rocío N. Ramos-Bernal [1], René Vázquez-Jiménez [1,2,*], Claudia A. Cantú-Ramírez [3], Antonio Alarcón-Paredes [4], Gustavo A. Alonso-Silverio [4], Adrián G. Bruzón [5], Fátima Arrogante-Funes [6], Fidel Martín-González [7], Carlos J. Novillo [2] and Patricia Arrogante-Funes [5]

1    Cuerpo Académico UAGro CA-93 Riesgos Naturales y Geotecnología, FI, Universidad Autónoma de Guerrero, Av. Lázaro Cárdenas s/n, CU, Chilpancingo 39070, GE, Mexico; rnramos@uagro.mx
2    Research Group on Technologies for Landscape Analysis and Diagnosis (TADAT), Department of Chemical and Environmental Technology, ESCET, Rey Juan Carlos University, C/Tulipán s/n, Móstoles, 28933 Madrid, Spain; carlos.novillo@urjc.es
3    Ingeniería para la Innovación y Desarrollo Tecnológico, Universidad Autónoma de Guerrero, Av/Lázaro Cárdenas s/n, CU, Chilpancingo 39070, GE, Mexico; acantu@uagro.mx
4    Cuerpo Académico UAGro CA-178 Desarrollo Tecnológico Aplicado, Universidad Autónoma de Guerrero, Av/Lázaro Cárdenas s/n, CU, Chilpancingo 39070, GE, Mexico; aalarcon@uagro.mx (A.A.-P.); gsilverio@uagro.mx (G.A.A.-S.)
5    Department of Chemical and Environmental Technology, ESCET, Rey Juan Carlos University, C/Tulipán s/n, Móstoles, 28933 Madrid, Spain; adrian.bruzon@urjc.es (A.G.B.); patricia.arrogante@urjc.es (P.A.-F.)
6    Grupo de Investigación en Teledetección Ambiental, Unidad Docente de Geografía, Geología y Medio Ambiente, Área de Geografía, Universidad de Alcalá, Filosofía y Letras, C/Colegios 2, 28801 Alcalá de Henares, Spain; fatima.arrogante@uah.es
7    Área de Geología, ESCET, Universidad Rey Juan Carlos, c/Tulipán s/n, Móstoles, 28933 Madrid, Spain; fidel.martin@urjc.es
*    Correspondence: rvazquez@uagro.mx

**Abstract:** Landslides are recognized as high-impact natural hazards in different regions around the world; therefore, they are extensively researched by experts. Landslide inventories are essential to identify areas that are likely to be affected in the future, thereby enabling interventions to prevent loss of life. Today, through combined approaches, such as remote sensing and machine learning techniques, it is possible to apply algorithms that use data derived from satellite images to produce landslide inventories. This work presents the performance of five machine learning methods—k-nearest neighbor (KNN), stochastic gradient descendent (SGD), support vector machine radial basis function (SVM RBF Kernel), support vector machine (SVM linear kernel), and AdaBoost—in landslide detection in a zone of the state of Guerrero in southern Mexico, using continuous change maps and primary landslide factors, such as slope angle, terrain orientation (aspect), and lithology, as inputs. The models were trained with 2/3 of ground truth samples of 671 slidden/non-slidden polygons. The obtained inventory maps were evaluated with the remaining 1/3 of ground truth samples by generating a confusion matrix and applying the Kappa concordance coefficient, accuracy, precision, recall, and F1 score as evaluation metrics, as well as omission and commission errors. According to the results, the AdaBoost classifier reached greater spatial and statistical coherence than the other implemented methods. The best input layer combination for detection was the continuous change maps obtained by the linear regression and image differencing detection methods, together with the slope angle, aspect, and lithology conditioning factors.

**Keywords:** landslides; continuous change maps; conditioning factors; machine learning

## 1. Introduction

Significant efforts have been made worldwide to collect, record, and analyze data on the occurrence and impacts of natural disasters [1]. The development of natural disaster databases is essential since they facilitate the evaluation of their natural and social impacts and the vulnerability of regions at different scales, which can be used to design and implement risk management or land-use planning by the corresponding authorities [2]. Geographic databases that contain information on landslides, including inventories and thematic data, represent a powerful tool for local, regional, national, or continental management and organization [3].

Inventory maps are essential components of a geographical landslide database. They provide historical information on past landslide events, including their location, type, and triggers, such as heavy rain, rapid thaw, or an earthquake. Inventories also include statistics on the frequency of slope failures and provide relevant information to build models of landslide susceptibility or landslide risk [4–10].

Most studies on landslide susceptibility, landslide risk, or landslide vulnerability start from the data of previous events integrated into landslide inventory maps, which are useful for evaluating the predictive capacity of the applied methods. Van Westen et al. (2008) [11] suggest that "a comprehensive landslide inventory is essential to be able to quantify both landslide hazard and risk." Thus, inventory generation is a crucial stage in this type of study and can be improved by applying techniques such as image analysis, historical research, or the integration of field sampling [12], which requires more significant resources and raises some difficulties in the integration of historical information, often due to a lack of available information [13].

The approaches applied in integrating landslide inventory maps have evolved with technological advances that have allowed access to satellite images with better spatial, radiometric, and temporal resolutions. The inventories of landslides integrated from visual inspections and mapping of landslides on photographs or images, complemented through field studies, have been helpful in the integration of maps of hazards and susceptibility to landslides [14]. According to Harp et al. [15], in the early 1960s, the first earthquake-induced landslide inventory maps were made by using aerial photographs; however, access to satellite images has allowed the integration of landslide inventories applying semi-automatic methods [14,16], detection of changes and image fusion [17].

Automatic change detection methods and learning algorithms that use images have been applied to integrate landslide inventory maps. Ramos-Bernal et al. (2018) [18] obtained a global landslide inventory map from the integration of three landslide inventory maps generated by applying three automatic change detection methods and two thresholding methods; the global map showed a mean omission error of 7.97%, a mean commission error of 6.79%, and a Kappa concordance coefficient of 84.15%. Chen et al. (2018) [19] obtained a landslide inventory map of Shenzhen, Zhouqu County, and Beichuan County, China, by applying a deep convolution neural network (DCNN) to extract areas where drastic alterations had occurred, followed by the application of spatial–temporal context learning (STCL) to identify landslide areas. The map obtained a commission error below 17.6% and reached a quality percentage higher than 61.1% for landslide areas. Hacıefendioğlu et al. (2021) [20] applied Resnet-50, VGG-19, Inception-V3, and Xception deep learning techniques to automatically detect regional landslides. They used receiver operating curves (ROCs) and F1 scores applicable to the characteristics of landslide detection to evaluate the experimental results. The results obtained show that both Resnet-50 and VGG-19 had a success rate greater than 90%.

### Conditioning Factors of Terrain Stability

Integrating a landslide inventory map over a large area means limiting the number of factors to consider because it is impossible to effectively control all of them due to either data availability or economic or time constraints. For this reason, it is vital to understand

how some individual factors influence the landslide process and thus decide whether to incorporate them into the analysis.

The spatiotemporal distribution of landslides is conditioned by factors that may be due to site characteristics or external agents. The factors can be many and varied; they interact in a complex way and can be classified into conditioning factors and triggers according to their characteristics. Inventories of landslides analyzed with the factors that conditioned or triggered them can generate helpful information for spatial models.

Various factors were identified in previous works on landslide susceptibility or landslide risk. Certain common factors were considered in many works, such as slope, aspect, geological lineaments, and lithology [12,19,21–31]. In addition, the number and types of considered factors always depend on the characteristics and conditions of the study area. Some of the most common factors among the analyzed works are slope angle, aspect, and lithology, which are included in this study mainly based on the analysis of the individual contribution of each of the mentioned factors, carried out through graphs of response curves and the jackknife test, developed by Ramos-Bernal [32].

The slope is the angle between the plane of the Earth's surface and the horizontal plane. The effects of gravity that determine the water flows and the materials removed vary depending on the slope (on higher slopes, the gravity and the speed of materials are more significant). This combination causes erosion, water and material transportation, and the induction of stress on the slopes, thus increasing the likelihood of landslides.

According to previous works, the slope angle is one of the most important factors to consider in landslide studies. It determines the amount of kinetic and potential energy that a stable mass can store or generate when unstable. Sites with low slope angles are considered more stable and safer from the occurrence of landslides; in other words, the possibility of a hillside failure tends to increase with a higher slope angle. A long hillside can include sections that may be affected by movements from the upper sections; thus, the steepness of the terrain indicates whether an area is prone to sliding. Several studies have confirmed the hypothesis that landslides occur more frequently on steeper surfaces [33–35]. According to Alcántara-Ayala (2000) [36], slopes equal to or greater than 5° favor the occurrence of landslides.

The terrain orientation represents the direction of exposure at a point, defined as the angle between the geographic north (azimuth) and the projection on the horizontal plane of the vector normal to the surface at a given point, with values of −1° to represent flat and continuous areas and values from 0° to 360° depending on the orientation [32]. The terrain orientation provides us with information about the insolation of a slope; it can also be interpreted in terms of vegetation. Depending on the rain front, the orientation influences the amount of rain and the direction in which it impacts the slope during precipitation events.

According to Guzzetti et al. (1999) [37], Cevik and Topal (2003) [38], Lee et al. (2004) [39], and Chen et al. (2015) [31], there is an important relationship between the terrain orientation and the occurrence of landslides; thus, it is often considered in studies aiming to model landslides. DeGraff and Romesburg (1980) [40] pointed out that the orientation reflects the structural and organic conditions of a slope because it includes fault planes and climatic factors. Marston et al. (1998) [41] observed that exposed soil on south-facing slopes in the Himalayas was subject to several wetting and drying cycles, which increased landslide activity. Other researchers, such as Chen et al. (2015) [31] and Rawat et al. (2015) [30], reported an association between landslides and the terrain orientation and classified them to identify the class range in which a more significant number of landslides occurred. The results of their work indicate that, in their study area, landslides tended to occur in orientations from southeast to east.

Lithology includes the structure, composition, texture, and other attributes that influence the physicochemical behavior of rocks and sediments, which together determine cohesion, internal friction angle, permeability, and susceptibility to weathering, whether physical or chemical.

Lithological and structural variations often lead to a difference in the strength and permeability of rocks and soils. Some works have recognized lithology as one of the most critical factors in slope sliding. Different lithological units present different degrees of landslide susceptibility; some soft lithologies weather more easily and are more prone to sliding than other hard lithologies [42–45].

The availability of remote sensing data is helpful to produce landslide inventory maps. Spectral data combined with specific factors that influence the landslide process can be indispensable for detecting landslides at a specific moment.

Given the above context, this work aims to integrate landslide inventory maps by applying supervised machine learning (ML) classification algorithms to continuous change maps derived from change detection techniques and maps of conditioning factors that influence land instability. This study referenced data from 2013 because extraordinary hydrometeorological events occurred that year: tropical depression No. 13 in the Pacific Ocean and two simultaneous hurricanes, Manuel in the Pacific and Ingrid in the Gulf of Mexico, caused significant flooding and landslides across the coast of the State of Guerrero.

## 2. Study Area

The study area covers 3300 km$^2$ in the central zone in the State of Guerrero in México; it consists of a mountainous region with elevations ranging from 280 m to 3540 m above mean sea level and slopes greater than 40° (Figure 1).

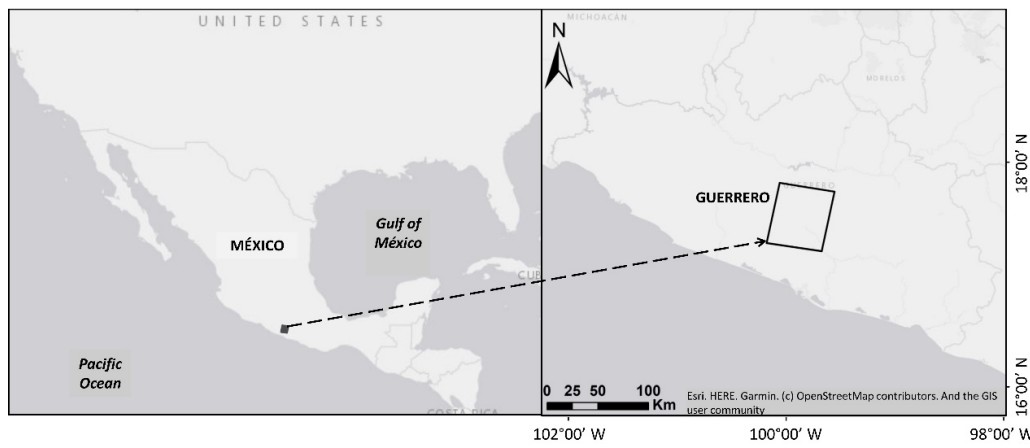

**Figure 1.** Study area. The central region of the State of Guerrero in México.

The precipitation records of the Mexican Meteorological Service (SMN) for 2013 indicate that the average varied from a minimum of 800 mm to a maximum of 2100 mm from June to September. The area was covered by 74.8% forest (coniferous, mesophilic, and mixed), 14.1% deciduous forest, 7.8% agricultural areas, 3.2% induced vegetation, and 0.1% human settlements and urban areas [46].

The area is physiographically located in the Sierra Madre del Sur [47]. It has various metamorphic rock compositions consisting of schists and gneisses of biotite and quartzite, deposited limestone outcrops, metavolcanic rocks with sedimentary influence, siltstones, sandstones, conglomerates, and carbonate rocks. Rhyolitic rocks are also found because of Oligocene–Miocene volcanism. The youngest rocks correspond to alluvial deposits present in riverbanks and channels [48,49].

The study area is interesting for its geographical location, its topographic and geo-logical conditions, and the occurrence of hydrometeorological phenomena in recent years, triggering massive landslides that have affected the population and the infrastructure in their communities. September 2013 was particularly notable, as this is when tropical depression No. 13 in the Pacific Ocean and subsequent simultaneous hurricanes—Manuel in the Pacific and Ingrid in the Gulf of Mexico—caused major landslides off the coast of the State of Guerrero. The landslide in the La Pintada community in Atoyac de Álvarez

stands out as particularly devastating (Figure 2a), causing 70 fatalities, 379 victims, and 20 damaged buildings [18].

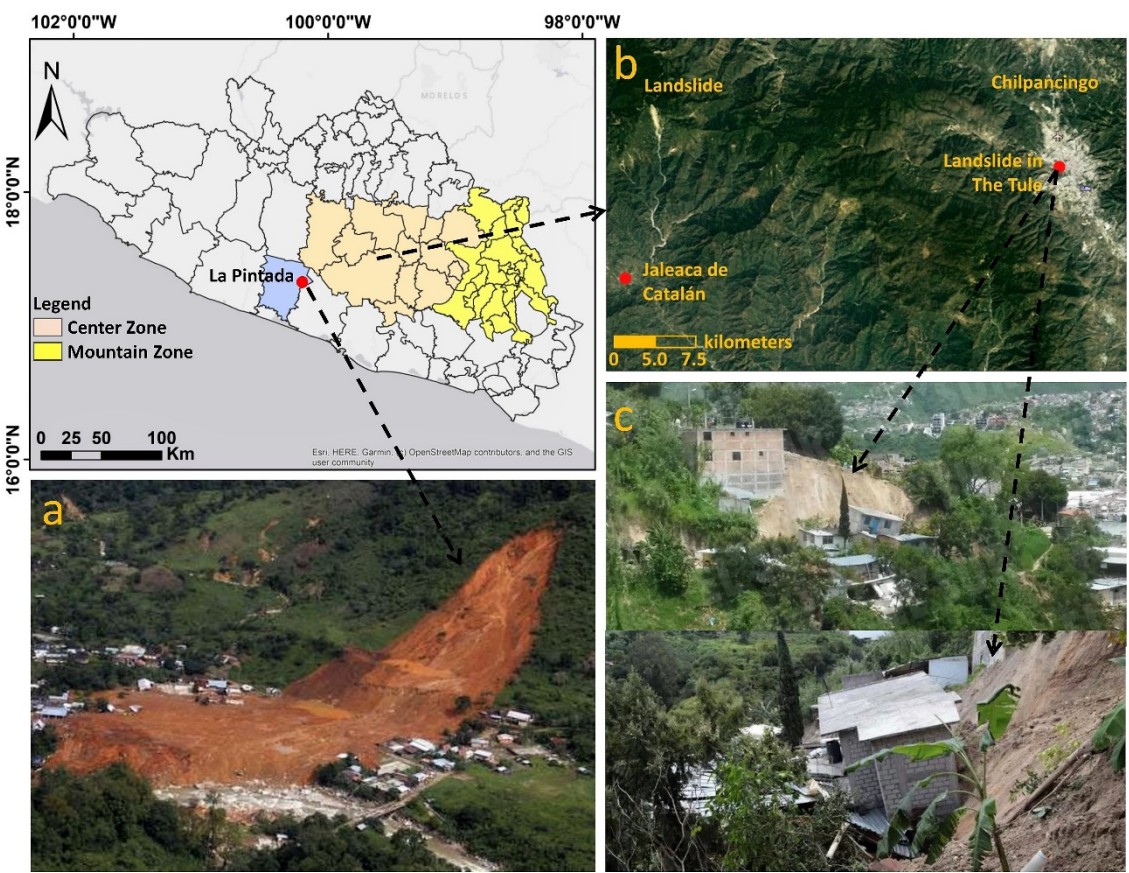

**Figure 2.** Landslides registered in the State of Guerrero. (**a**) Massive landslide in La Pintada community. Source: Adaptation https://www.jornada.com.mx/2013/09/24/ciencias/a03a1cie (accessed on 25 May 2021). (**b**) Landslide in Jaleaca de Catalán and (**c**) landslide in El Tule ravine in Chilpancingo, the capital city of the State of Guerrero. Source: Adaptation in Google Earth and El Sur, 2017. https://suracapulco.mx/aun-no-reubican-a-20-familias-que-perdieron-su-casa-tras-un-deslizamiento-de-tierra-en-la-capital/(accessed on 25 May 2021).

The mountain region (yellow in Figure 2), located on the west of Guerrero State, was one of the areas most affected by heavy rains in 2013. Twenty-nine fatalities were confirmed, and the Ministry of Agrarian, Territorial and Urban Development (SEDATU) reported that 4351 homes in the mountain region were affected (some experienced fissures or subsidence, and others were destroyed), but local sources in the region assert that an additional 2988 homes were damaged [50,51]. The central region was also affected by the occurrence of landslides; one of them is particularly notable for its size (2.5 × 0.5 km), which occurred approximately 14 km north of the community of Jaleaca de Catalán and 33 km northwest of Chilpancingo, the state capital (Figure 2b).

In September 2017, due to intense rainfall, Chilpancingo, the state capital, was affected by a landslide that led to the collapse of ten houses near the El Tule ravine (Figure 2c), located between the El Mirador and Obrera neighborhoods [52].

There is a significant concentration of inhabitants in the study area. According to the 2010 population and housing census (the closest census year to the occurrence of landslides in the study area), 187 localities with 15,230 dwellings were occupied by 59,098 inhabitants [53].

## 3. Materials and Methods

The methodology developed for the integration of landslide inventories consists of four stages:

1.  Dataset. The first step consists of acquiring primary data: ASTER images, Digital Elevation Model (DEM), geological-mining charts, aerial photographs, and field data. Derived maps were generated from Principal Component 1 (PC1) in principal component analysis (PCA), normalized difference vegetation index (NDVI), cloud masks, slope angle (S), aspect (A), lithology (L), and ground truth (GT) samples. Dinamica EGO 5 was used to generate the NDVI images; the PCA images were obtained using ArcMap 10.3.

2.  Automatic change detection. At this stage, two change detection methods were applied, linear regression (LR) and image differencing (Diff), to the maps derived from the ASTER images. Three maps of continuous change were generated: LR applied to PC1 (LR-PC1), LR applied to NDVI (LR-NDVI), and Diff applied to PC1 (Diff-PC1). This process was performed with Dinamica EGO 5.

3.  Supervised classification. In this stage, the k-nearest-neighbor (KNN), stochastic gradient descent (SGD), support vector machine (SVM), and AdaBoost classifiers were applied using the previously obtained continuous change maps (LR-PC1, LR-NDVI, and Diff-PC1) and the factors of slope stability (S, A, and L) considered. The process was repeated and complemented by incorporating the factors into the classification, one by one, and combining them. All classification algorithms, so as the accuracy evaluation metrics, were run using the scikit-learn version 0.22.1 for Python 3.6.5 programming language.

4.  Accuracy assessment. In this stage, the inventory maps obtained by supervised classification were evaluated by confusion matrices, omission and commission errors, and metrics such as the Kappa concordance coefficient (k), accuracy (ACC), precision, recall, and F1 score. Python's Georasters library version 0.5.20 was used to read the derived satellite images, including its metadata, and generate the maps after the classification stage.

The overall methodology used in the study is depicted in Figure 3.

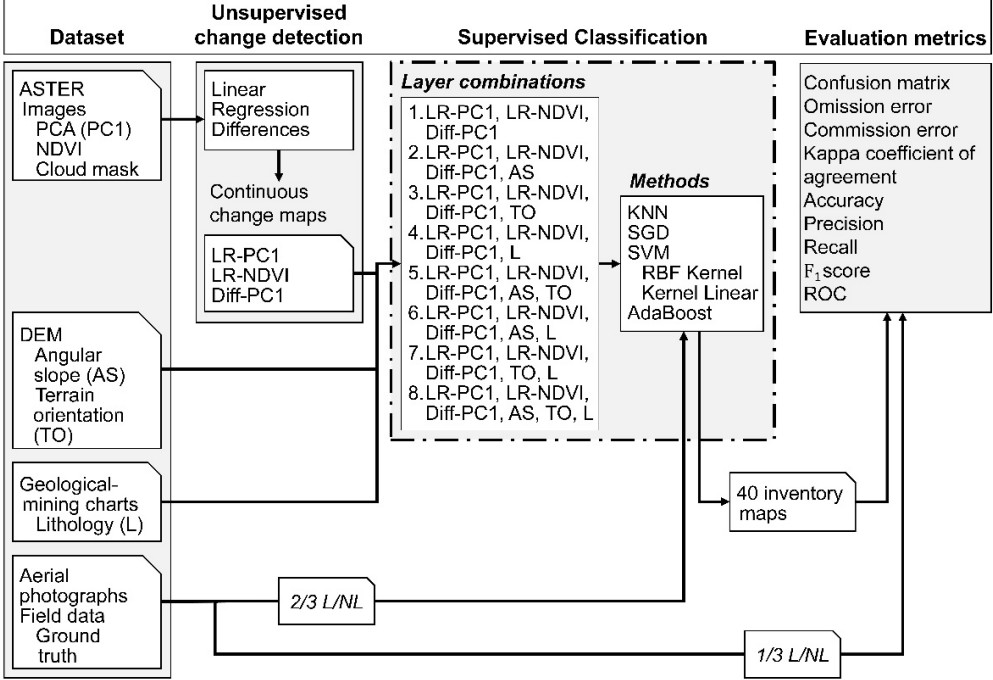

**Figure 3.** Flowchart: methodology overview.

### 3.1. Dataset

Two ASTER (level AST-L1T) images were used courtesy of the United States Geological Survey; the images cover the study area and were captured on 10 December 2012, and 13 December 2013 (before and after the extraordinary hydrometeorological event of September 2013). The green, red, and near-infrared bands were used with a spatial resolution of 15 m. The SCS + C method (Sun Canopy Sensor + Correction) was employed for topographic correction by slope class [54] to better characterize the diffuse irradiance; this approach is recommended for mountainous forest areas [55]. From these normalized images, new images were generated through principal component analysis transformation [56] and normalized difference vegetation indices [57,58] using a cloud mask and shadows. The derived images were used in the change detection stage.

From the topographic maps, courtesy of the National Institute of Statistics and Geography of Mexico (INEGI), a digital elevation model (DEM) of 15 m spatial resolution was obtained to generate the slope angle (S) and aspect (A) maps. The lithology map (L) was obtained from geological-mining charts courtesy of the Mexican Geological Service (SGM). These three maps were used as factors in the supervised classification stage (Figure 4).

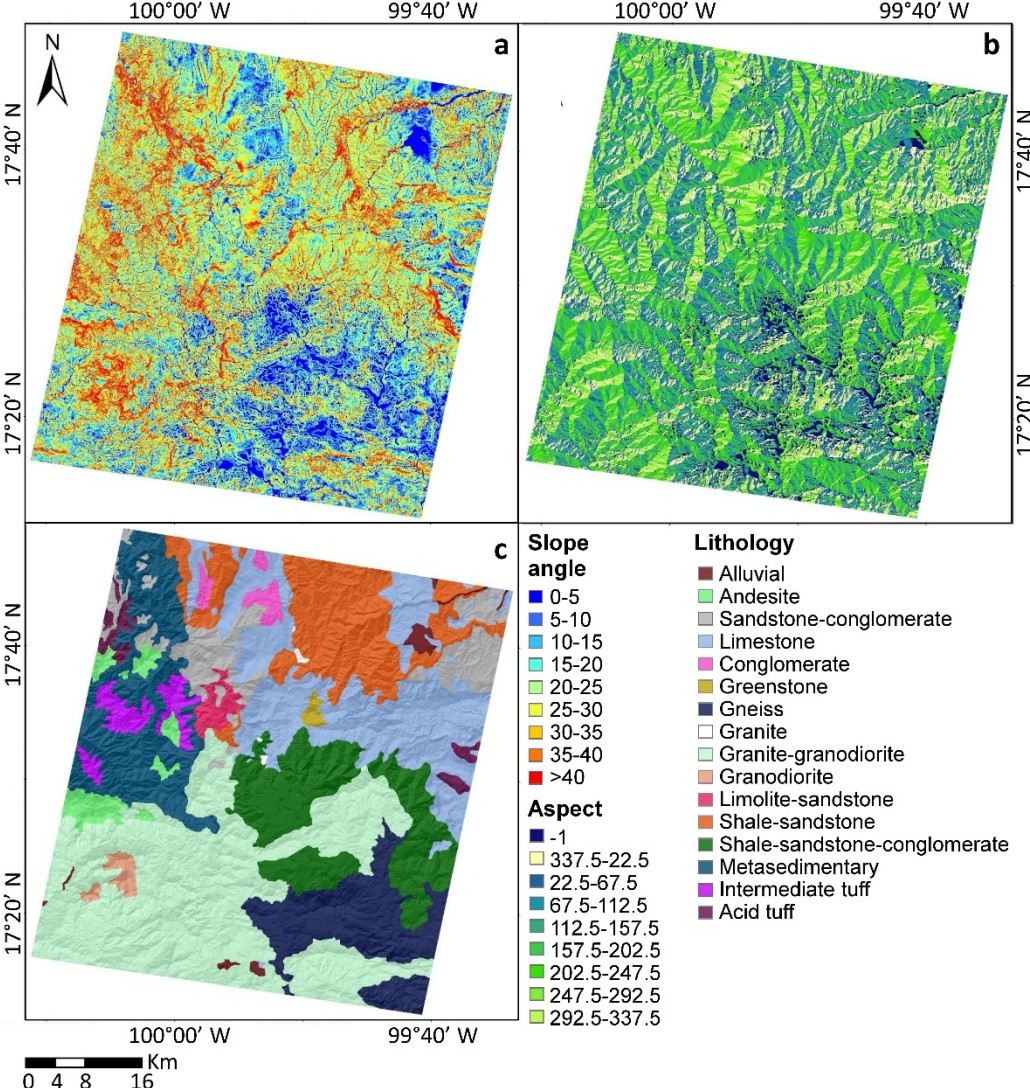

**Figure 4.** Explanatory factors of hillside land instability: (**a**) slope angle (S), (**b**) aspect (A), and (**c**) lithology (L).

A Google Earth image from 12 August 2014 was obtained (the closest available to the meteorological events of September 2013).

### 3.2. Ground Truth

The ground truth (GT) was integrated by sampling actual landslide and non-landslide zones and digitizing polygons on the image from 12 August 2014, on the Google Earth platform. When sampling the GT inventory, areas with surfaces greater than 450 m$^2$ (two ASTER pixels) were given priority to characterize small landslides in greater detail.

Due to the time difference between the image used for integrating the GT and the date of the extraordinary rainfall (11 months), a validation was applied to ensure that the polygons identified in Google Earth existed in the 2013 ASTER image. As a result, 671 polygons were digitized and converted to raster format, representing 32,462 pixels of the study area: 592 polygons (15,266 pixels) correspond to landslides, and 79 polygons (17,196 pixels) correspond to non-landsides (Figure 5). The polygons were included within the landslide areas to obtain more reliable results.

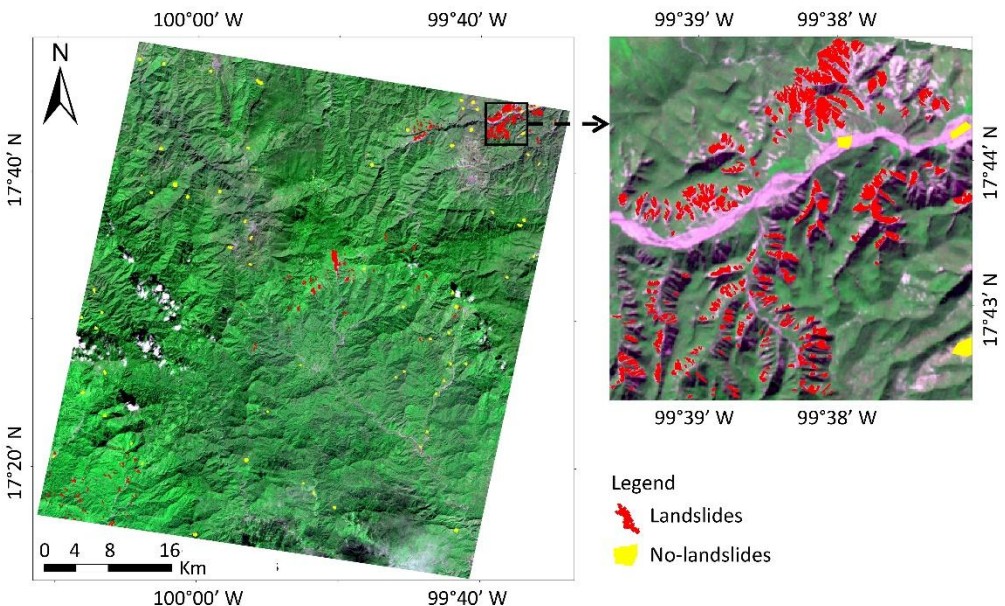

**Figure 5.** Ground truth (GT) sampling.

An evaluation of the GT was carried out through field visits to some areas. Due to issues of access and unsafe conditions in the study area, not all digitized polygons were field verified. Field verification was performed on 321 polygons (53.3%) in safe areas close to the largest cities and near the most important roads. The GT must be accurate and timely; in this study, the GT refers to sites in which actual significant landslides were previously identified and sites considered unaltered or stable within the study area, determined through Google Earth images and partially validated by field check.

Among the areas identified as non-landslides in the GT sampling, different land use and land cover areas were included, such as forests, low deciduous forests, agriculture, bodies of water, and human settlements, to evaluate only the capacity of the model to identify landslides.

The GT samples were randomly divided into two parts, as follows:

- 2/3 of the GT samples (21,640 pixels) were used for the training stage of the classification models to identify landslides/non-landslides;
- 1/3 of the GT samples (10,822 pixels) were reserved for assessing the accuracy of the classifier methods in detecting landslides.

### 3.3. Automatic Change Detection

Three maps of continuous change were generated and used as data layers in the supervised classifier to integrate landslide inventories. The continuous change maps were obtained from Principal Component 1 (PC1). The normalized difference vegetation index

(NDVI) maps were derived from the normalized ASTER images and the application of linear regression and image differencing as change detection methods, which are described below.

### 3.3.1. Linear Regression

The linear regression (LR) change detection method assumes that the pixel values (Y) of the final-date image $f_2$ result from a linear function of the pixel values (X) of the initial-date image $f_1$. Thus, it is possible to perform a regression from $Y_{I,J}^K(f_2)$ to $X_{I,J}^K(f_1)$ by least squares [59–61] to obtain the parameter gradient *m* and Y-intercept *b* of the regression line and generate a model equation in the form $Y' = mX + b$.

In this case, we applied LR to the values of PC1 and NDVI images to obtain a new image, $Y_{I,J}'^K, (f_2 - f_1)$, for each index that corresponds to the expected values generated by the prediction model. With the expected and actual PC1 and NDVI values of *f2*, residual images of the PC1 and NDVI values can be calculated by the following equation:

$$R_{i,j}^k = Y_{I,J}'^K, (f_2 - f_1) - Y_{i,j}^k(f_2) \tag{1}$$

where $R_{i,j}^k$ is the residual pixel value of line *i* and column *j* for band *k*; $Y_{I,J}'^K, (f_2 - f_1)$ is the image created by the prediction model; and $Y_{i,j}^k(f_2)$ is the image with the actual PC1 or NDVI values of the final date.

The pixel values in the expected image obtained by the predicted model will be the same as the actual pixel values of the final date only if no changes were recorded during the analyzed period (residual = 0). On the other hand, a residual with a value other than zero indicates a change. The magnitude of the value indicates the intensity of the change (Figure 6b,c).

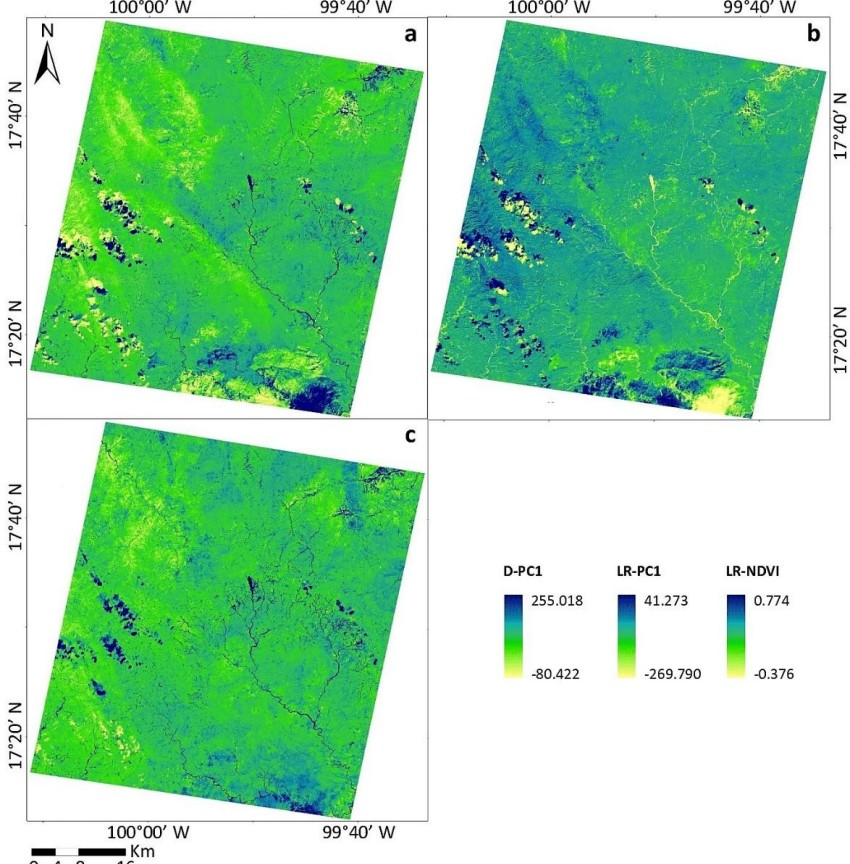

**Figure 6.** Continuous change maps obtained by automatic change detection methods. (**a**) Diff-PC1, (**b**) LR-PC1, and (**c**) LR-NDVI.

### 3.3.2. Image Differencing

This change detection method obtains a simple difference between the pixel values of two images from different dates by subtracting the initial-date image $f_1$ from the final-date image $f_2$, pixel-by-pixel, shown in Equation (2).

$$Diff_{i,j} = Y_{i,j}f_2 - X_{i,j}f_1 \tag{2}$$

where $Diff_{i,j}$ represents the continuous change image, $Y_{i,j}f_2$ is the value of the later date, and $X_{i,j}f_1$ is the value of the earlier date. For this case, the values correspond to PC1 and NDVI, and the images are from 13 December 2013, and 10 Decemeber 2012.

The values obtained by the change prediction model will be equal to zero if no changes were registered during the analyzed period (residual = 0). On the other hand, if there were changes, they will be reflected in the corresponding residual value, and the magnitude indicates the intensity of the change (Figure 6a).

### 3.4. Supervised Classification

Landslide detection was performed using supervised learning models, specifically classification algorithms. The classification algorithms work in two main stages:

- First, the algorithm undergoes a learning process by generating knowledge from the association between known input and output data;
- Second, the corresponding output values are estimated based on new input data.

These estimated values are compared with the GT samples to measure the algorithm's performance through different evaluation metrics.

In this study, the input data are defined by the data from the layers obtained in the previous automatic change detection stage, and they are grouped into one of two possible output classes: landslide or non-landslide zones.

The use of machine learning models to solve spatial modeling problems involving natural hazards, such as landslide assessments, has increased in recent years. These models have some advantages over conventional models, e.g., the possibility of adjusting their internal structure based on experimental data, working with big data, and predicting categorical factors to provide cost-effective, fast, and accurate models [62].

As mentioned before, this work aims to identify landslide-affected areas by applying k-nearest-neighbor (KNN), stochastic gradient descent (SGD), support vector machine (radial basis function kernel and linear kernel), and AdaBoost machine learning models to a combination of continuous change images (LR-PC1, LR-NDVI, and Diff-PC1) during the analyzed period and conditioning factors (S, A, and L), which are used as inputs in the supervised classification. The machine learning algorithms used were implemented in Python programming language, using the Scikit-Learn library in version 0.21.3.

Classifiers

K-nearest neighbor is an algorithm that compares unknown input data to previously known data and assesses their similarity. This comparison is carried out by computing the distance (typically Euclidean) between the pixel values of the conditioning factors in the area to be classified and the pixel values of all known zones. The next step is to identify k zones that are most similar (those with the shortest distance) to the zone to be classified, and finally, the most frequent class among the k-nearest zones is assigned. Pixels with similar conditioning factors will have a smaller distance between them, compared to pixels with significantly different conditioning factors. The k parameter is commonly an odd positive integer to avoid a tie between classes [63].

Stochastic gradient descent is a simple but efficient approach to adjust convex loss functions in linear classifiers or regressors, such as SVM (linear) and logistic regression. Strictly speaking, stochastic gradient descent is more like an optimization technique and does not correspond to a specific family of machine learning models, and thus, it is used only for training a model. The main advantage of SGD is its efficiency since it is linear

in the number of training data, so it is less complex than other algorithms and also has a lower computational cost. For example, if there is a matrix of size (n, p), the training step has a cost *O(knp)*, where *k* is the number of iterations (epochs), and *p* is the average number of non-zero attributes per sample [64].

Support vector machine (SVM), which is a supervised machine learning classifier that arose from the theory of non-parametric statistical learning [65] and the principle of structural risk [62,66,67], is widely used in the generation of cartography for landslide detection [62,68–71]. SVM uses training data to transform the initial input space to a higher feature space to generate optimal hyperplanes between two different classes and maximize the margin between those classes. These hyperplanes are known as support vectors and make the classifier more robust to noise, as they take advantage of the maximum margin between the landslide and non-landslide classes [65,72]. To achieve this separation, a suitable kernel function must be selected [62]. The kernels most frequently used are linear, polynomial, radial basis function, and sigmoid. The SVM classifier was applied in two variants in this work, with one using the radial basis function kernel (RBF kernel) and the other using the linear kernel.

AdaBoost was proposed by Freund and Schapire (1997) [73] and is one of the most widely used boosting algorithms [74]. It involves the application of an adaptive resampling technique and can improve predictions because it controls bias and variance [75–77]. Boosting is a methodology that combines the performance of several weak classifiers, such as logistic regression [78], functional tree [79], or neural networks [80], for the training dataset and generates a robust classifier. The AdaBoost technique can be described in three steps:

1. A subset of training data is randomly generated from the original training dataset, each of which is assigned equal weights;
2. The misclassified data are given greater weight, whereas correctly classified data still have the same weight;
3. The first step, followed by a normalization process, is repeated, and a new training subset is generated. This process is performed repeatedly and ends when a predefined stop condition is reached, obtaining a lower error value than expected [75,81].

AdaBoost is sensitive to atypical values and can affect the accuracy of the classifier. By combining several models, the boosting method can predict the landslide/non-landslide classes better than using a single model. In this work, a decision tree was used as the weak classifier to implement AdaBoost.

### 3.5. Experimental Description

In the supervised classification, two groups of images were applied:

1. The spectral images analyzed correspond to the best continuous change images produced by the change detection method in the previous stage, as described in Ramos-Bernal et al. (2015) [16] and Ramos-Bernal et al. (2018) [18], corresponding to LR for PC1 and NDVI images, and Diff for PC1 images. Those images were selected based on relevant differences in sites where landslides occurred during the analyzed period.
2. The combinations of maps with the S, A and L factors were included in the landslide detection process, generating 40 landslide and non-landslide maps.

The combinations of input maps in each classification step are shown in Table 1. For example, combination C1 includes LR-PC1, LR-NDVI, and Diff-PC1; combination C2 includes the three previous images and the slope angle.

**Table 1.** Image combinations of input data in supervised classification.

| Images | Input Classification | | | | | | | |
|---|---|---|---|---|---|---|---|---|
| LR-PC1 | * | * | * | * | * | * | * | * |
| LR-NDVI | * | * | * | * | * | * | * | * |
| Diff-PC1 | * | * | * | * | * | * | * | * |
| Slope angle (S) | | * | | | * | * | | * |
| Aspect (A) | | | * | | * | | * | * |
| Lithology (L) | | | | * | | * | * | * |
| Combination | C1 | C2 | C3 | C4 | C5 | C6 | C7 | C8 |

* image/factor included in the combination.

All classification methods were run using their default parameters. Table 2 describes the parameters used to train classifiers. The classification process used 2/3 of the GT samples to train the models.

**Table 2.** Parameters used in the training of classifiers.

| Classifiers | Parameters |
|---|---|
| KNN | N_neighbors = 5, weights = "uniform", algorithm = "auto", leaf_size = 30, $p$ = 2, metric = "minkowski", mertric_params = None, n_jobs = None, Kwargs |
| SGD | Loss = "hinge", penalty = "l2", alpha = 0.0001, l1_ratio = 0.15, fit_intercept = True, max_iter = 1000, tol = 0.001, shuffle = True, verbose = 0, epsilon = 0.1, n_jobs = _None, random_state = 0, learning_rate = "optimal", eta0 = 0.0, power_t = 0.5, early_stopping = False, validation_fraction = 0.1, n_iter_no_changes = 5, class_weight = None, warm_start = False, average = False |
| SVM RBF kernel | C = 10, kernel = "rbf", degree = 3, gamma = "auto_deprecate", coef0 = 0.0, shrinking = True, probability = False, tol = 0.001, cache_size = 200, class_weight = None, verbose = False, max_iter = −1, decision_fuction_shape = "ovr", random_state = 0. |
| SVM linear kernel | Penalty = "l2", loss = "squared_hinge", dual = True, tol = 0.0001, C = 1.0, multi_class = "ovr", fit_intercept = True, intercept_scaling = 1, class_weight = None, verbose = 0, random_state = None, max_iter = 1000 |
| AdaBoost | Base_estimator = None, n_estimators = 50, learning_rate = 1.0, algorithm = "SAMME.R", random_state = 0 |

*3.6. Accuracy Assessment*

One-third of the GT subsample (10,822 pixels) was reserved for the evaluation stage. A cross-tabulation was performed between the GT subsample and each thematic map resulting from the classification, generating confusion matrices, omission and commission errors, and the Kappa concordance index [82] to quantify the concordance between the observed GT map and the randomly expected map. The Kappa concordance coefficient attempts to define the degree of adjustment only by the precision of categorization, regardless of random causes [56,83].

In addition, accuracy (ACC), precision, recall, and F1 score were obtained and used as evaluation metrics. Accuracy corresponds to the rate of correctly classified output values: it is calculated by summing the number of values correctly predicted as landslides and those correctly predicted as non-landslides, and then dividing the resulting value by the total number of GT values. Precision evaluates how many of the values predicted by the classifier to be landslides correspond to actual landslides. High precision means that the model has a high probability of generating a correct landslide classification. On the other hand, recall measures the number of landslides correctly identified by the classifier relative to the total number of landslides in the GT. Since the precision and recall methods evaluate different aspects of the classifier, the F1 score combines these two metrics to obtain the harmonic mean, where a score of 0 is the worst value and 1 is the best score. Table 3 shows the equations of the evaluation metrics.

**Table 3.** Evaluation metrics.

| Method | Equations |
|---|---|
| Kappa concordance coefficient | $k = \left(n\sum_{i=1,n} X_{ii} - \sum_{i=1,n} X_{i+} X_{+i}\right) / \left(n^2 - \sum_{i=1,n} X_{i+} X_{+i}\right)$ <br> $k$ is the Kappa coefficient of agreement, $n$ is the sample size, $X_{ii}$ is the observed agreement, and $X_{i+} X_{+i}$ is the expected agreement in each category $i$. |
| Accuracy | $ACC = \frac{TP+TN}{TP+TN+FP+FN}$ <br> $TN$ corresponds to true-negative pixels; $FP$ represents False-Positive pixels; $TP$ represents true-positive pixels, and $FN$ represents False-Negative pixels. |
| Precision | $Precision = \frac{TP}{TP+FP}$ <br> $FP$ represents False-Positive pixels, and $TP$ represents True-Positive pixels. |
| Recall | $Recall = \frac{TP}{TP+FN}$ <br> $TP$ represents True-Positive pixels, and $FN$ represents false-negative pixels. |
| $F_1$ score | $F_1\ score = \frac{2 \cdot Precision \cdot Recall}{Precision + Recall}$ |

## 4. Results

### 4.1. Supervised Classification

The proposed methodology described in Figure 3 was applied to the eight-layer combination of satellite-derived images (Table 1). The resulting landslide/non-landslide maps are depicted in Figures 7–11.

Figure 12 presents a close-up visualization of the eight maps generated by the AdaBoost classifier for a more detailed analysis.

Figure 13 shows the same region but displays the maps generated by all methods in this study (KNN, SGD, SVM RBF kernel, SVM linear kernel, and AdaBoost) when applying combination 1 (LR-PC1, LR-NDVI, and Diff-PC1) in landslide detection.

### 4.2. Accuracy Assessment

As described in the proposed methodology, the maps resulting from each classification method applied to each combination were evaluated by comparing them to the GT samples reserved for evaluation. The resulting values of the evaluation metrics applied in the comparison are shown in Table 4.

As defined in Table 3, the Kappa concordance coefficient was computed using a confusion matrix resulting from a cross-comparison between the GT samples reserved for the evaluation step and the maps resulting from each classification method and each combination, considering the commission and omission errors for each analyzed category (landslide/non-landslide). In contrast to the evaluation metrics, the Kappa index integrates both successes and errors in each category into a single index, making it more suitable for evaluation purposes in this study. Figure 14 depicts the Kappa concordance coefficient for each method and combination.

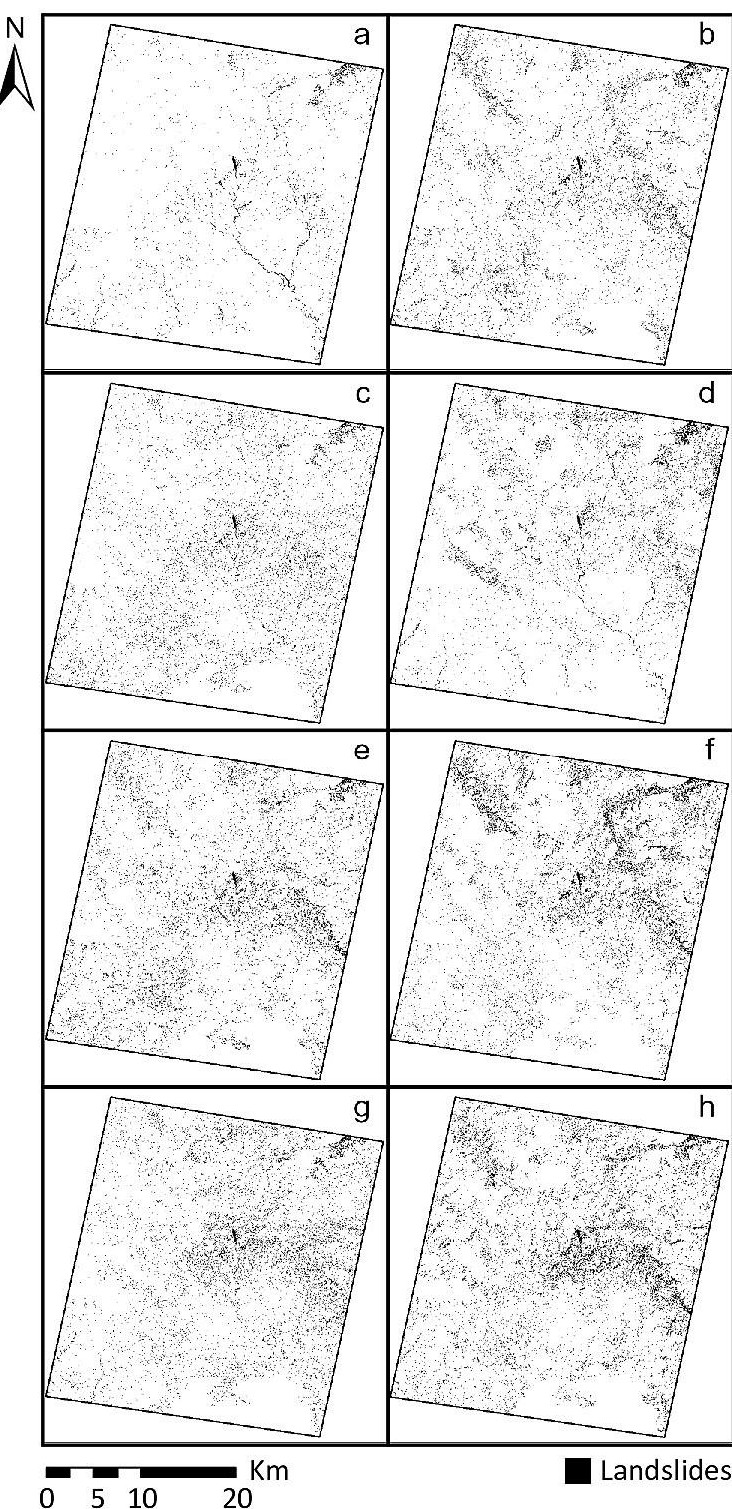

**Figure 7.** Landslide inventories generated by KNN using combinations (**a**) C1, (**b**) C2, (**c**) C3, (**d**) C4, (**e**) C5, (**f**) C6, (**g**) C7, and (**h**) C8.

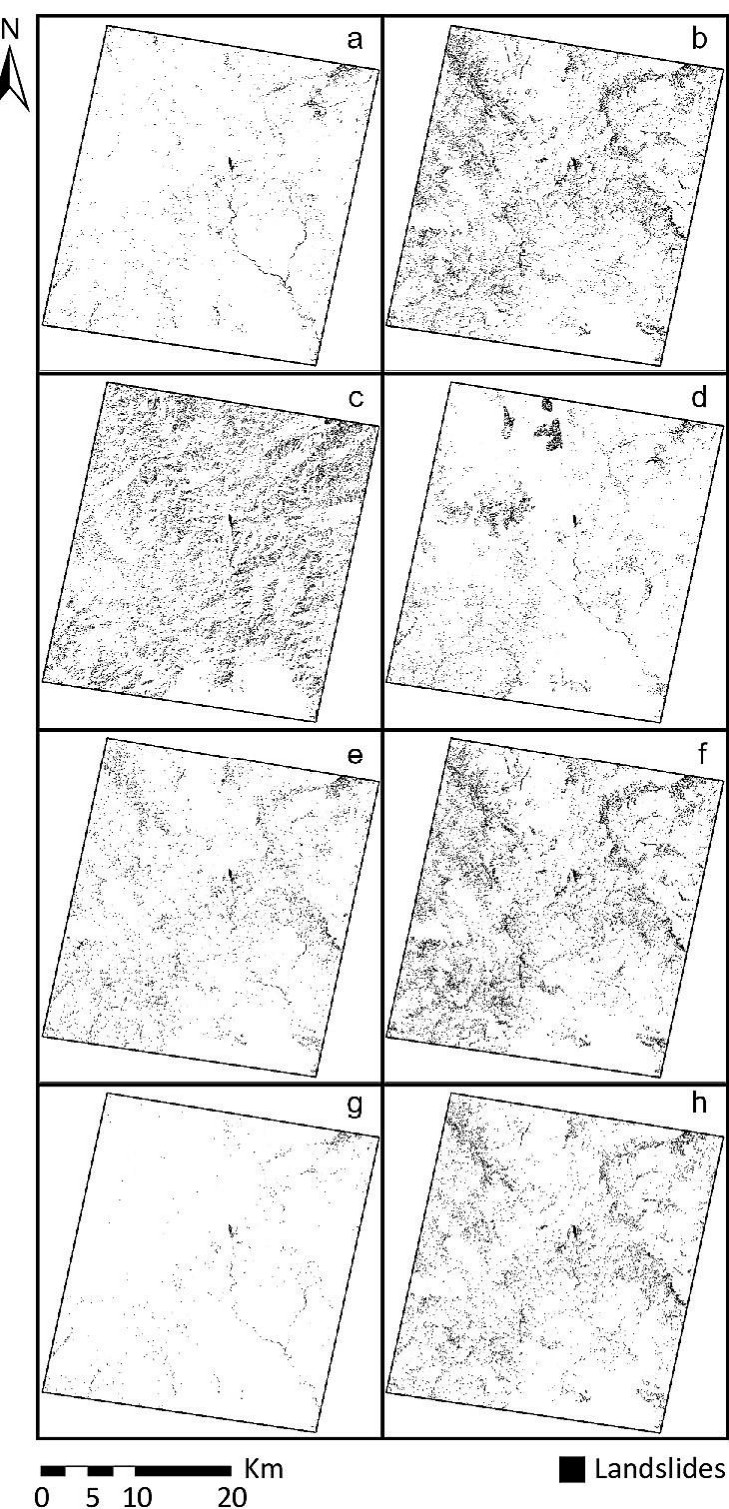

**Figure 8.** Landslide inventories generated by SGD using combinations (**a**) C1, (**b**) C2, (**c**) C3, (**d**) C4, (**e**) C5, (**f**) C6, (**g**) C7, and (**h**) C8.

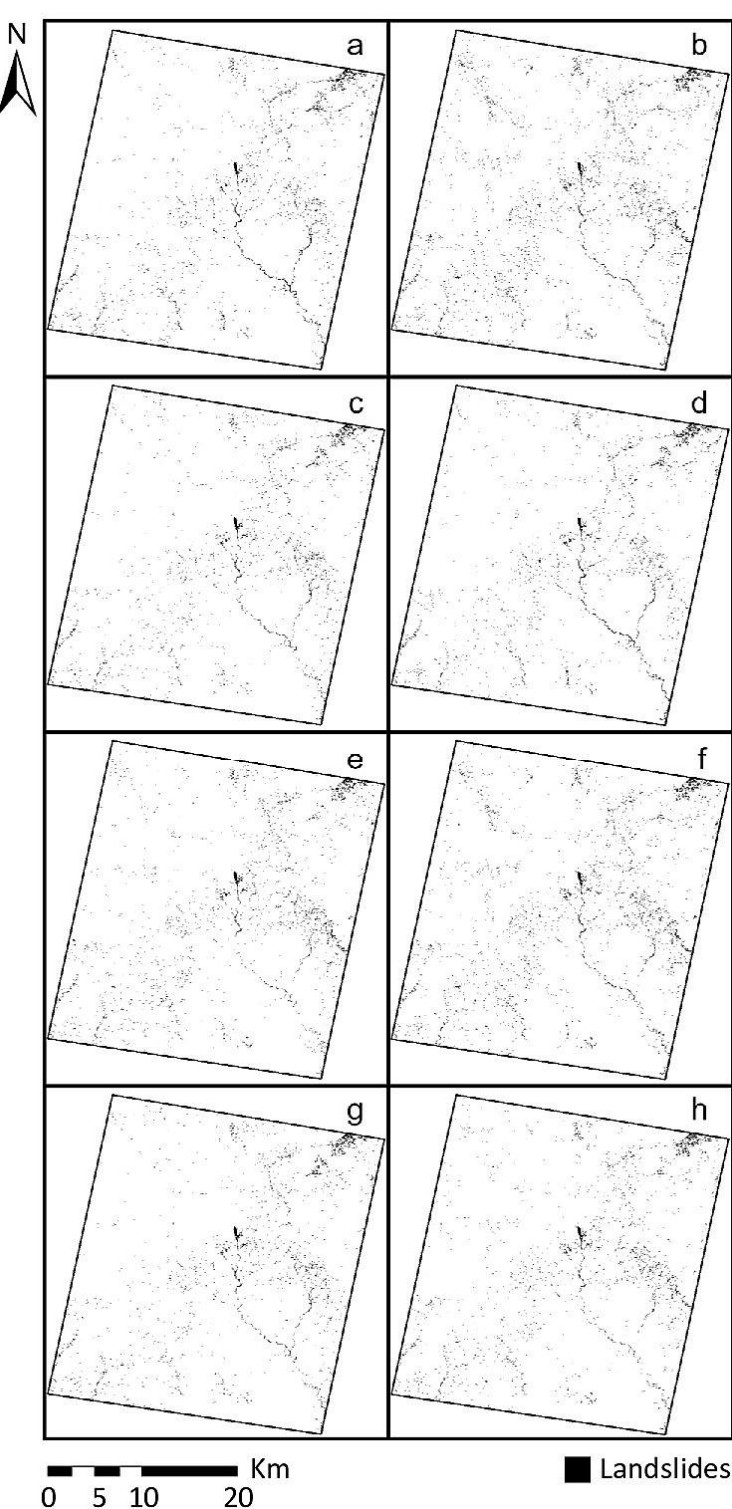

**Figure 9.** Landslide inventories generated by SVM with RBF Kernel using combinations (**a**) C1, (**b**) C2, (**c**) C3, (**d**) C4, (**e**) C5, (**f**) C6, (**g**) C7, and (**h**) C8.

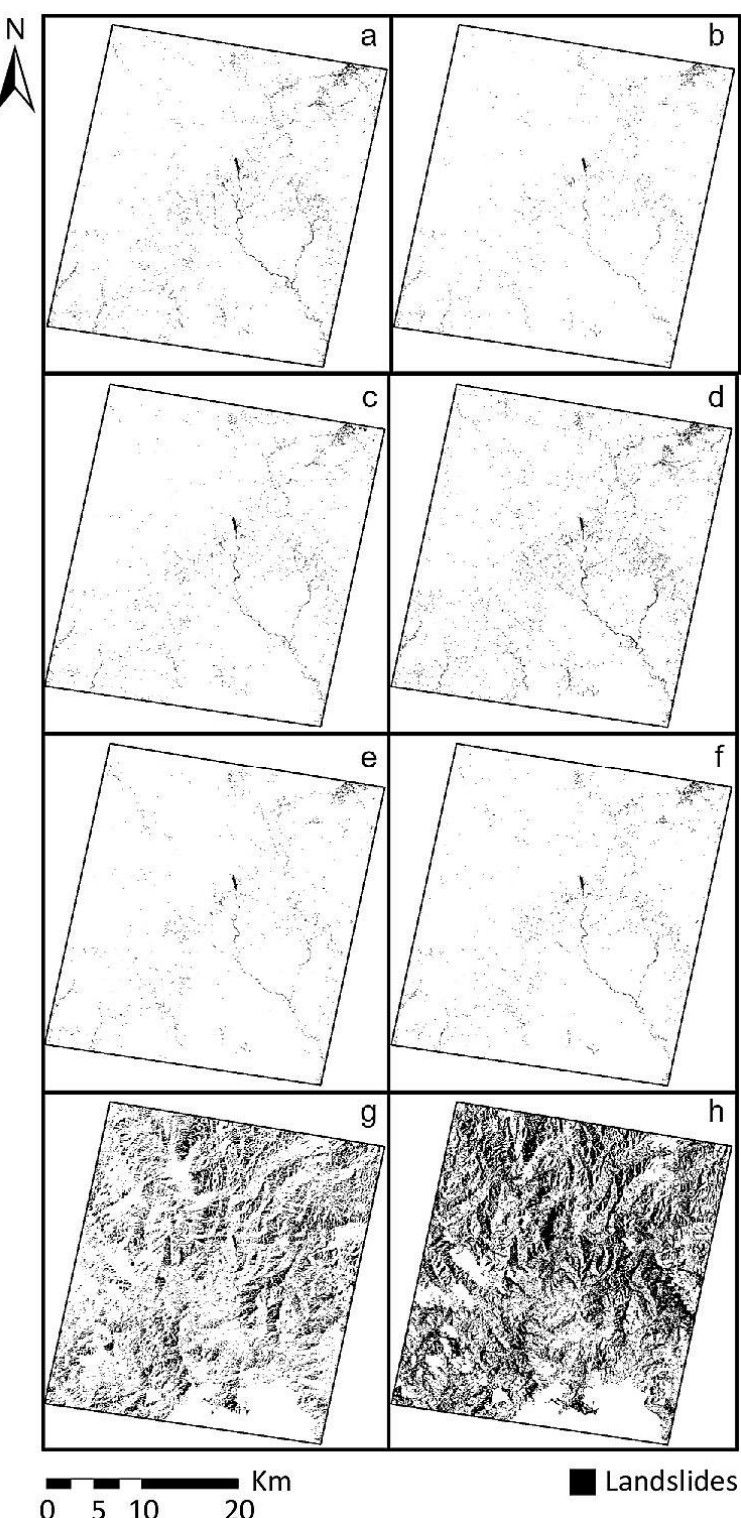

**Figure 10.** Landslide inventories generated by SVM with linear kernel using combinations (**a**) C1, (**b**) C2, (**c**) C3, (**d**) C4, (**e**) C5, (**f**) C6, (**g**) C7, and (**h**) C8.

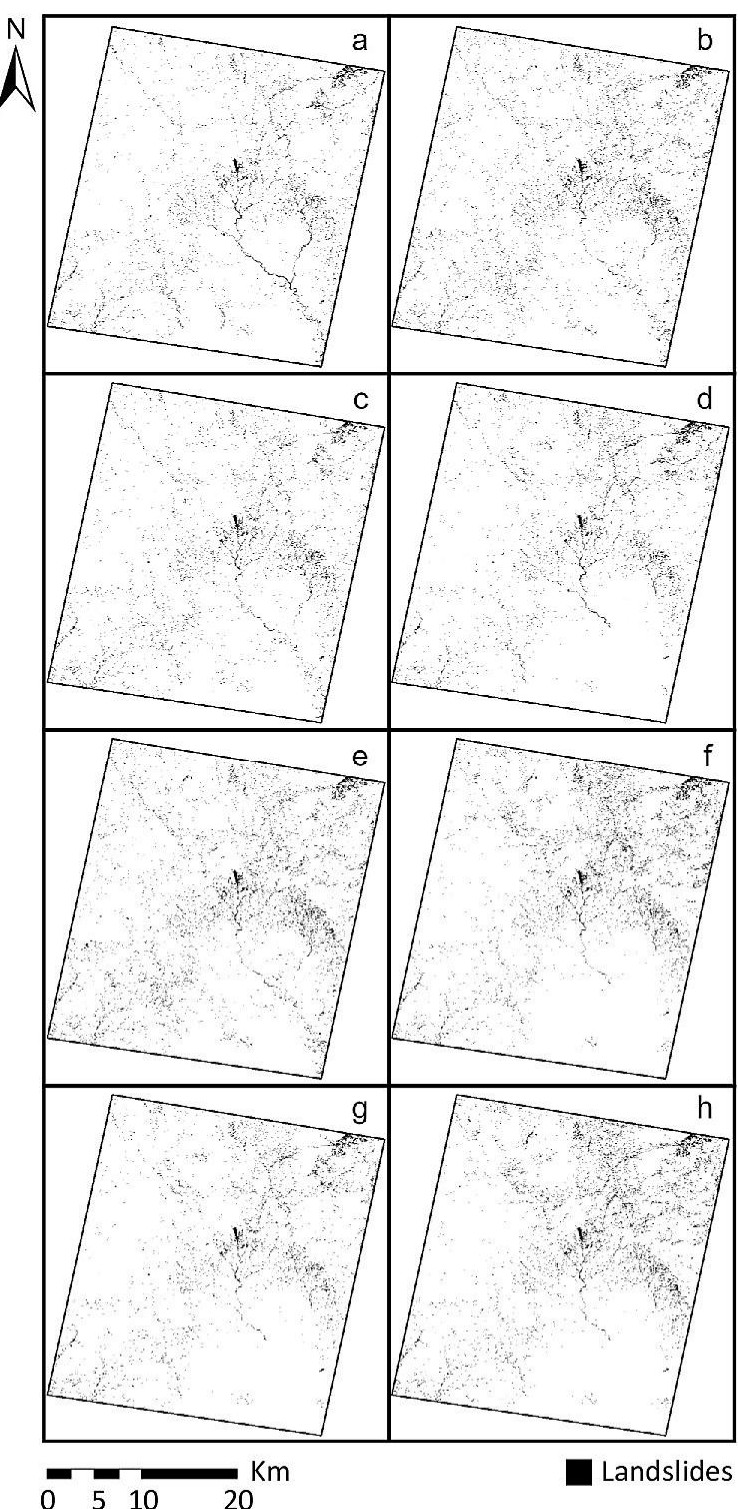

**Figure 11.** Landslide inventories generated by AdaBoost using combinations (**a**) C1, (**b**) C2, (**c**) C3, (**d**) C4, (**e**) C5, (**f**) C6, (**g**) C7, and (**h**) C8.

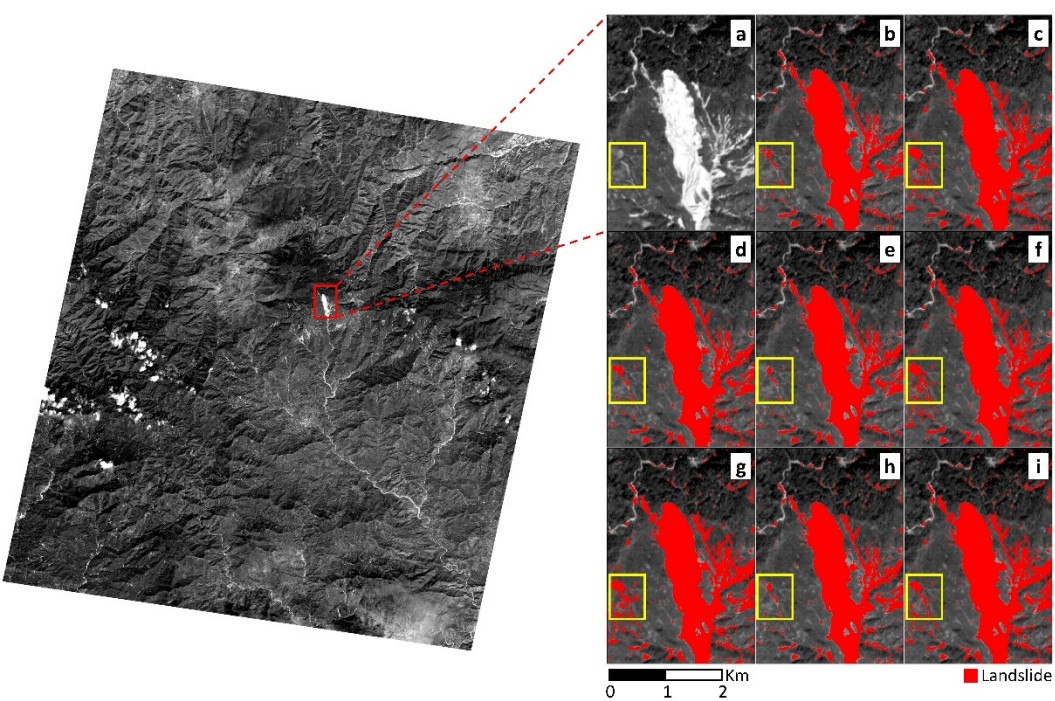

**Figure 12.** Close-up of the maps obtained by the AdaBoost classifier on the area where the largest landslide occurred: (**a**) Red band. Results of landslide detection using combinations (**b**) C1, (**c**) C2, (**d**) C3, (**e**) C4, (**f**) C5, (**g**) C6, (**h**) C7, and (**i**) C8.

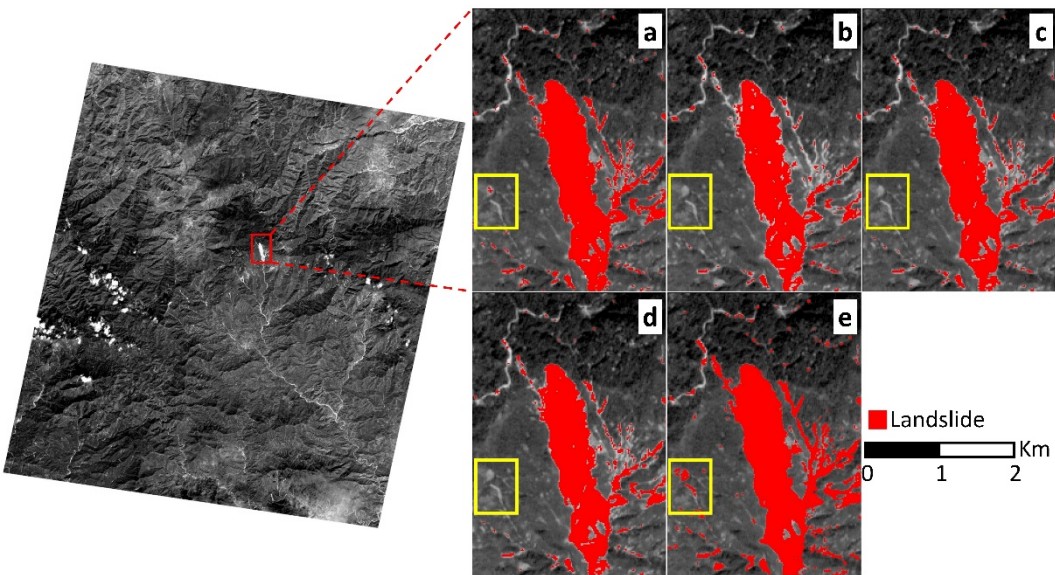

**Figure 13.** Details on landslide region classification when applying (**a**) KNN, (**b**) SGD, (**c**) SVM RBF kernel, (**d**) SVM linear kernel, and (**e**) AdaBoost using combination C1 (LR-PC1, LR-NDVI, and Diff-PC1).

**Table 4.** Performance evaluation of landslide/non-landslide maps.

| Method | Combination | Precision | Recall | F1 Score | Accuracy | Kappa | Omission Error | Commission Error |
|---|---|---|---|---|---|---|---|---|
| KNN | C1 | 0.94 | 0.90 | 0.92 | 0.93 | 0.850 | 7.324 | 7.569 |
| | C2 | 0.97 | 0.93 | 0.95 | 0.96 | 0.911 | 4.274 | 4.527 |
| | C3 | 0.97 | 0.92 | 0.94 | 0.94 | 0.888 | 5.372 | 5.700 |
| | C4 | 0.96 | 0.93 | 0.95 | 0.95 | 0.899 | 4.927 | 5.092 |
| | C5 | 0.98 | 0.94 | 0.96 | 0.96 | 0.927 | 3.480 | 3.714 |
| | C6 | 0.98 | 0.95 | 0.97 | 0.97 | 0.937 | 3.043 | 3.252 |
| | C7 | 0.97 | 0.93 | 0.95 | 0.95 | 0.907 | 4.479 | 4.718 |
| | C8 | 0.99 | 0.95 | 0.97 | 0.97 | 0.944 | 2.658 | 2.860 |
| SGD | C1 | 0.91 | 0.82 | 0.86 | 0.88 | 0.751 | 11.884 | 12.657 |
| | C2 | 0.93 | 0.92 | 0.92 | 0.93 | 0.857 | 7.083 | 7.165 |
| | C3 | 0.85 | 0.87 | 0.86 | 0.87 | 0.732 | 13.414 | 13.366 |
| | C4 | 0.89 | 0.84 | 0.87 | 0.88 | 0.751 | 12.253 | 12.567 |
| | C5 | 0.96 | 0.9 | 0.93 | 0.94 | 0.873 | 6.075 | 6.470 |
| | C6 | 0.93 | 0.91 | 0.92 | 0.92 | 0.846 | 7.650 | 7.712 |
| | C7 | 0.96 | 0.62 | 0.75 | 0.81 | 0.606 | 15.232 | 20.211 |
| | C8 | 0.95 | 0.91 | 0.93 | 0.94 | 0.875 | 6.083 | 6.324 |
| SVM RBF kernel | C1 | 0.93 | 0.87 | 0.9 | 0.91 | 0.818 | 8.823 | 9.232 |
| | C2 | 0.96 | 0.9 | 0.93 | 0.94 | 0.870 | 6.268 | 6.628 |
| | C3 | 0.95 | 0.88 | 0.91 | 0.92 | 0.839 | 7.708 | 8.193 |
| | C4 | 0.93 | 0.87 | 0.9 | 0.91 | 0.817 | 8.853 | 9.272 |
| | C5 | 0.96 | 0.9 | 0.93 | 0.94 | 0.876 | 5.891 | 6.322 |
| | C6 | 0.96 | 0.9 | 0.93 | 0.93 | 0.867 | 6.411 | 6.791 |
| | C7 | 0.95 | 0.88 | 0.91 | 0.92 | 0.840 | 7.684 | 8.164 |
| | C8 | 0.96 | 0.9 | 0.93 | 0.94 | 0.872 | 6.077 | 6.532 |
| SVM linear kernel | C1 | 0.93 | 0.87 | 0.9 | 0.91 | 0.817 | 8.830 | 9.268 |
| | C2 | 0.98 | 0.78 | 0.87 | 0.88 | 0.767 | 9.718 | 12.020 |
| | C3 | 0.95 | 0.82 | 0.88 | 0.9 | 0.789 | 9.539 | 10.803 |
| | C4 | 0.90 | 0.91 | 0.91 | 0.91 | 0.821 | 8.952 | 8.912 |
| | C5 | 0.97 | 0.76 | 0.85 | 0.88 | 0.750 | 10.412 | 12.890 |
| | C6 | 0.97 | 0.85 | 0.9 | 0.91 | 0.825 | 7.944 | 9.010 |
| | C7 | 0.84 | 0.86 | 0.85 | 0.85 | 0.709 | 14.550 | 14.472 |
| | C8 | 0.73 | 0.97 | 0.83 | 0.81 | 0.627 | 15.866 | 18.157 |
| AdaBoost | C1 | 0.93 | 0.93 | 0.93 | 0.93 | 0.861 | 6.934 | 6.945 |
| | C2 | 0.98 | 0.97 | 0.98 | 0.98 | 0.956 | 2.163 | 2.208 |
| | C3 | 0.97 | 0.94 | 0.96 | 0.96 | 0.922 | 3.806 | 3.969 |
| | C4 | 0.95 | 0.96 | 0.95 | 0.96 | 0.914 | 4.317 | 4.294 |
| | C5 | 0.98 | 0.97 | 0.98 | 0.98 | 0.956 | 2.162 | 2.208 |
| | C6 | 0.98 | 0.98 | 0.98 | 0.98 | 0.965 | 1.767 | 1.772 |
| | C7 | 0.98 | 0.97 | 0.97 | 0.97 | 0.947 | 2.609 | 2.651 |
| | C8 | 0.98 | 0.98 | 0.98 | 0.98 | 0.962 | 1.920 | 1.894 |

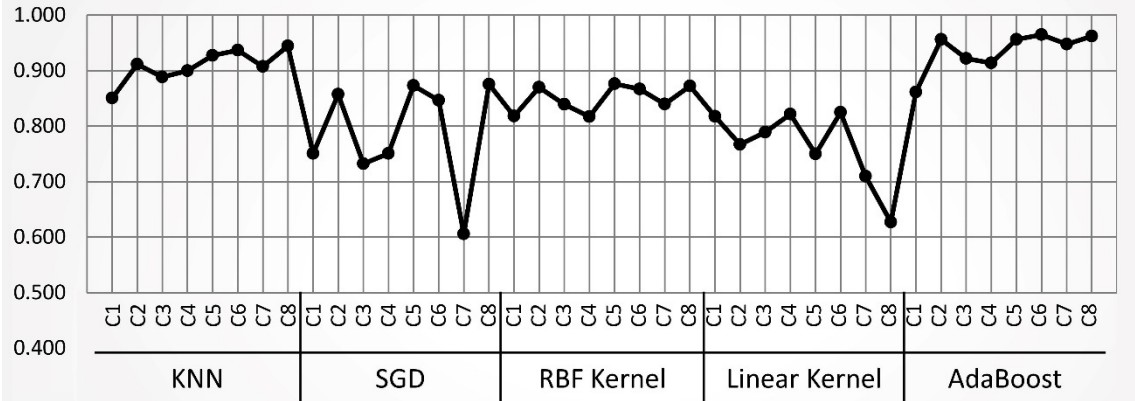

**Figure 14.** Kappa concordance coefficient for each method and combination employed in this study.

## 5. Discussion

### 5.1. Visual Analysis

At first glance, the resulting maps (Figures 7–11) seem to match the areas where landslides are predicted. However, we also observe differences, with some maps, including many more landslide pixels than others (Table 5).

**Table 5.** Landslide detection ratio by classification method for each layer combination. The mean ratio and the standard deviation are also included.

| Method | C1 | C2 | C3 | C4 | C5 | C6 | C7 | C8 | Std. Dev. | Mean |
|---|---|---|---|---|---|---|---|---|---|---|
| KNN | 1.14 | 2.83 | 2.68 | 2.73 | 3.46 | 4.39 | 3.40 | 4.81 | 1.13 | 3.18 |
| SGD | 0.78 | 4.47 | 5.81 | 1.87 | 1.81 | 4.99 | 0.29 | 2.85 | 2.03 | 2.86 |
| SVM RBF kernel | 0.82 | 1.04 | 0.73 | 0.81 | 0.82 | 0.97 | 0.73 | 0.79 | 0.11 | 0.84 |
| SVM linear kernel | 0.80 | 0.38 | 0.63 | 1.2 | 0.44 | 0.56 | 11.76 | 25.09 | 8.96 | 5.11 |
| AdaBoost | 1.56 | 2.00 | 1.59 | 1.60 | 2.12 | 2.13 | 1.38 | 2.14 | 0.31 | 1.82 |

As shown in Table 4, some maps have a high tendency to detect landslides (SVM linear kernel+C7 and SVM linear kernel+C8), whereas other maps have a lower landslide detection ratio (SVM RBF kernel+C3 and SVM RBF kernel+C7). Furthermore, the results suggest that methods with lower standard deviations will generate more coherent maps, namely, SVM RBF kernel and AdaBoost; this statement can be confirmed by inspecting their corresponding maps, as shown in Figures 9 and 11.

According to Figure 11 and the landslide ratio results reported in Table 4, the maps obtained by the AdaBoost classifier (Figure 12) have the highest visual coherence.

A close-up of the region in the study area where the largest landslide was triggered (2.5 × 0.5 km) is depicted in Figure 12. An initial visual inspection seems to suggest that the eight maps correctly identify this landslide. However, some differences can be observed in regions where more minor landslides occurred, such as the region outlined by the yellow bounding box, in which a method may detect more extensive zones (Figure 12c,f,g) or smaller areas (Figure 12b,d,e,h) depending on the combination of continuous change maps and the inclusion of conditioning factors.

At a glance, the maps in Figure 13 may look similar; however, differences are detected upon detailed inspection. In the map generated by AdaBoost, the pixels for landslides are better identified throughout the entire zone (Figure 13e). On the other hand, the SGD-generated map fails to identify some landslide zones, as can be seen in the largest landslide and other minor ones (Figure 13b). Moreover, the maps obtained by methods other than AdaBoost, such as SGD, SVM RBF kernel, and SVM linear kernel, do not detect some landslides in the yellow-marked region (Figure 13b–d), and the KNN map correctly identifies only a small zone of the landslides in the same area (Figure 13a).

As described above, the AdaBoost method outperforms the other classifiers in categorizing landslide zones; this conclusion is based on two principal findings: it fully covers the largest landslide and detects minor landslides omitted by the other classification methods.

### 5.2. Accuracy Assessment

According to the evaluation metrics shown in Table 4, for most of the maps, the values of precision, recall, F1 score, and accuracy exceed 0.80, which are considered acceptable. Analyzing the behavior of classification methods individually, AdaBoost shows more consistent results across the metrics with a standard deviation of 0.018, followed by KNN with 0.02, SVM RBF kernel with 0.026, SVM linear kernel with 0.06, and, finally, SGD with 0.068; thus, SGD has the most variance in its results.

Analyzing the metrics only for the maps obtained by the AdaBoost method but for different combinations, combination C8 (composed of LR-PC1, LR-NDVI, Diff-PC1, S, A,

and L) and combination C6 (composed of LR-PC1, LR-NDVI, Diff-PC1, S, and L) have the highest values, i.e., 0.98 for all analyzed metrics, representing the best results among the applied methods and the combinations of inputs.

Figure 14 shows the values of the Kappa concordance coefficient. The highest values were obtained by the AdaBoost classifier, with an average of 0.94, using combinations C6 (layers LR-PC1, LR-NDVI, Diff-PC1, S, and L) and C8 (layers LR-PC1, LR-NDVI, Diff-PC1, S, A, and L), with values of 0.965 and 0.962, respectively. Similarly, the AdaBoost method exhibits, on average, the lowest commission (3.21) and omission (3.24) errors among classifiers, averaging 3.23 in total. In addition, combinations C6 and C8 have the best average errors: 1.77 and 1.91, respectively.

Among the other classifiers, two methods obtained a Kappa index above 0.80: the KNN method has a Kappa average of 0.91 with average omission and commission errors of 4.44 and 4.68, and the SVM RBF kernel has a Kappa index of 0.85 with average omission and commission errors of 7.21 and 7.64. The remaining methods obtained a Kappa concordance coefficient below 0.80: SGD has an average Kappa value of 0.79 and omission and commission errors of 9.96 and 10.81, respectively, and SVM linear kernel has a Kappa value of 0.76 and higher omission and commission errors of 10.73 and 11.94.

Thus, the performance evaluation supports the previous visual examination. The maps resulting from SGD (combinations C7 and C3) and SVM linear kernel (combinations C7 and C8) obtained the lowest values of the Kappa concordance coefficient as well as the worst evaluation metrics (Table 4 and Figure 14). This result is validated by the visual incoherence visible in their corresponding maps (Figure 8c,g and Figure 10g,h).

*5.3. Conditioning Factor Analysis*

In this work, continuous change maps resulting from the change detection process between images from different dates were tested as alternative input layers for each algorithm (KNN, SGD, SVM RBF kernel, SVM linear kernel, and AdaBoost) in the classification stage. Thus, the landslide inventory maps depicted in Figures 7a, 8a, 9a, 10a and 11a, were obtained. In the visual inspection and performance evaluation, specific changes were observed when adding each conditioning factor or combining two or three factors with the obtained maps.

The AdaBoost and KNN methods show a general upward trend in the Kappa concordance coefficient (Figure 14) as conditioning factors are incorporated or combined, which is not observed for the rest of the classifiers, for which the Kappa index could increase or decrease when adding independent or combined factors. However, this upward trend is preserved by including the three continuous change images and the three conditioning factors, i.e., combination C8, into the classification step with the AdaBoost and KNN methods. Both cases generate excellent results, as shown in Table 5 and Figures 7h, 11h and 14.

The landslide inventory map obtained using the SGD method using combination C7 (LR-PC1, LR-NDVI, Diff-PC1, A, and L) only identifies 0.29% of the pixels classified as landslides (Figure 8g; Table 4), in contrast to the other inventory maps obtained with combinations C1–C6 and C8, as illustrated in Figure 8a–f,h.

As mentioned before, the slope angle (S) is a fundamental conditioning factor, and according to previous studies, it has a relevant impact on landslide occurrence. Focusing on the maps produced by the AdaBoost and KNN methods, the incorporation of the S factor in the classification step results in a significant increase in the Kappa values in comparison to those considering only the continuous change image; the values improve from 0.861 to 0.956 for AdaBoost and from 0.85 to 0.911 for KNN (Table 4). In general, when the S factor is included, e.g., combination C2, the Kappa concordance coefficient of the maps generated by the classification methods improves to values higher than 0.85, except for the map generated by the SVM linear kernel method, which is the only case in which the Kappa index drops (0.767) after the incorporation of the S factor. This exception may be due to the fact that the SVM linear kernel method with combination C2 also results in the lowest landslide detection ratio (only 0.38). Furthermore, the landslide inventory maps

obtained using the SVM linear kernel method applying combinations C7 and C8 have the largest number of pixels categorized as landslide (Figure 10g,h, Table 3).

As was previously mentioned, all classification methods were run using their default parameters. It is evident that a careful selection of parameters may lead the classification algorithms to better results regarding a specific region of interest. However, although there are some parameter selection methods (e.g., grid search or randomized search), we decided not to use any of them, as we wish to highlight this study as a generalized methodology for any other study area.

Some other works apply well-known deep learning architectures [19,20,84–86]. Chen et al. [19] and Hacıefendioğlu et al. [20] in their results report an average recall of 0.86 and 0.91, respectively. Lu et al. [84], Defang et al. [85] and Qin et al. [86] report an accuracy of 0.98, 0.98 and 0.88, respectively. As can be seen, their results are very similar to those achieved in this work (Table 4). However, their algorithms require higher computational costs than those presented in this study.

## 6. Conclusions

The established aim for this work was to integrate an inventory of landslides by applying supervised machine learning classification to continuous change maps derived from change detection techniques and conditioning factors of hillside terrain instability.

The results obtained in this study indicate that the AdaBoost algorithm achieved the highest precision indices and was thus, the most suitable. Therefore, this algorithm is recommended for generating landslide/non-landslide inventory maps in study areas with similar conditions to those in this work, using the methods and the conditioning factors described herein. Compared with previous work [18–20] that applied different methods and methodologies, the applied metrics and Kappa concordance coefficient were improved with the AdaBoost method, with values above 90% for these parameters and mean errors of omission and commission below 2.0%.

As a future line of research, we consider it appropriate to investigate some deep learning algorithms in order to establish more accurate comparisons with the method proposed in this study, and we will explore strategies to reduce the computational cost. For example, an original compact model can be used instead of a pre-trained architecture, or transfer learning can be used, which can improve the training stage of the categories of interest (in this case, landslides and non-landslides) by using the knowledge that the model in question learns for one task and applying it to training on a related task.

The sampling design and the integration of the ground truth are fundamental steps that define the capacity of the applied classification method, as feeding the classifier algorithm with the best combination of GT can maximize its ability to discriminate areas of landslides from those that remain stable.

Mapping information on landslide susceptibility is necessary for landslide-prone areas; it can support the establishment of early warning systems for residents in high-risk areas and can also assist in the management of events of this type by institutions and organizations responsible for safeguarding the public, allowing them to locate areas with greater vulnerability according to relevant factors and implement plans for the early detection or prevention of disasters in areas with higher priority.

This work can be complemented with the identification of evacuation routes, care centers, and shelters and their capacities, among other possible actions, all of which place a fundamental focus on acquiring knowledge of the territory through the application of geographic, computing, and information technologies.

**Author Contributions:** Conceptualization, R.N.R.-B. and R.V.-J.; methodology, R.N.R.-B. and R.V.-J.; software, C.A.C.-R. and A.G.B.; validation, P.A.-F., A.A.-P., G.A.A.-S. and A.G.B.; formal analysis, R.N.R.-B., R.V.-J., A.A.-P., G.A.A.-S., P.A.-F. and A.G.B.; investigation, R.N.R.-B., R.V.-J., C.A.C.-R., A.A.-P. and G.A.A.-S.; resources, C.A.C.-R., F.A.-F., F.M.-G., C.J.N. and A.G.B.; data curation, C.A.C.-R., F.A.-F., F.M.-G., C.J.N. and A.G.B.; writing—original draft preparation, R.N.R.-B. and R.V.-J.; writing—review and editing, R.N.R.-B., R.V.-J., A.A.-P., G.A.A.-S., P.A.-F. and A.G.B.; visualization,

F.A.-F., F.M.-G. and C.J.N.; supervision, R.N.R.-B. and R.V.-J.; project administration, P.A.-F. and A.G.B.; funding acquisition, P.A.-F. and A.G.B. All authors have read and agreed to the published version of the manuscript.

**Funding:** This research was funded by Rey Juan Carlos University under II call for financing of development cooperation projects: "Ciencia, tecnología y cooperación por el desarrollo sostenible: diseño y evaluación de una metodología de detección y predicción de deslizamientos de tierra basada en inteligencia artificial" and pre-doctoral contract program (Ref: PREDOC21-029) and by Spanish Ministry of Science and Innovation under the pre-doctoral contracts program (Ref: PRE2019-089208).

**Data Availability Statement:** Not applicable.

**Conflicts of Interest:** The authors declare no conflict of interest.

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
