# Peer review of "Evaluation of Conditioning Factors of Slope Instability and Continuous Change Maps in the Generation of Landslide Inventory Maps Using Machine Learning (ML) Algorithms"

_remotesensing, doi:10.3390/rs13224515_

Round 1
Reviewer 1 Report
- The title of section 1.1 on the bottom page 2 are incomprehensible, also the position of this section.
- The section 3.2, GT was integrated through a sampling by digitizing polygons on the image of August 12, 2014 on the Google Earth platform. The Google Earth image, passed 11 months after September, 2013, may be difficult to show actually landslides triggered by the tropical depression No. 13 in the Pacific Ocean and subsequence simultaneous hurricanes.
- Line 256-257, the GT inventory sampling, areas with surfaces greater than 450 m2 (two ASTER pixels). Here means, is 450m2 about two ASTER pixels? is it not 15 to 90 square meters per pixel of ASTER?
- Line 258, Six hundred seventy-one should be expressed 671 for being consistent with the following expression.
- 12-13 are the close-up to the maps, which location should be given in the whole study area.
- The accuracy of the 5 models may related with the parameters used in the training of classifiers, which shown in Table 2. So how to select the most optimal parameters is a key problem. The selection of parameters in Table 2 should be described in detail. Alternatively, it is suggested that the impact of parameters be explained in the discussion.
Author Response
Response to Reviewer 01
Manuscript ID: Remotesensing-1407105
Title: Evaluation of conditioning factors of slope instability and con-tinuous change maps in the generation of landslide inventory maps using Machine Learning (ML) algorithms
Journal: Remote sensing
We thank the anonymous reviewers for their valuable comments and suggestions that helped improve the manuscript's quality and clarity. A detailed point-by-point response to the reviewers’ comments (in blue) is provided in the following paragraphs.
REVIEWER 1
- The title of section 1.1 on the bottom page 2 are incomprehensible, also the position of this section.
The study has several dimensions, among which the evaluation of conditioning factors of slope instability stands out, as highlighted in the title and mentioned in the abstract; For this reason, the Introduction addresses general aspects about the landslide phenomenon, the importance of inventory maps, and the types of studies carried out to integrate landslide inventory maps.
Thus, within the Introduction, it is addressed as a subtopic: Conditioning factors of the terrain stability, where it is first attempted to address the role that conditioning and triggering factors play in the occurrence of landslides, then the great variety of factors that influence landslides is addressed. that make a comprehensive study complex; and finally we focus on the description of the factors that are addressed in the study.
When reviewing the subtopic, the authors found that it should start two paragraphs before addressing the relationship between the number of factors and the size of the study areas and their spatial distribution, being necessary to limit the number of factors in large areas.
- The section 3.2, GT was integrated through a sampling by digitizing polygons on the image of August 12, 2014 on the Google Earth platform. The Google Earth image, passed 11 months after September, 2013, may be difficult to show actually landslides triggered by the tropical depression No. 13 in the Pacific Ocean and subsequence simultaneous hurricanes.
First, the image of August 12, 2014, was taken because it was the next one available on the Google Earth platform after the date on which the extraordinary hydrometeorological phenomenon occurred in the study area, which caused innumerable landslides. As can be seen in the study, it effectively served to identify landslides, which were most likely generated by the extraordinary rains of 2013. However, during While digitizing the ground truth polygons on the 2014 Google Earth image, a validation was applied on the 2013 Aster image, ensuring that the polygons identified in Google Earth existed in the 2013 image. The use of the 2014 image and tools of the Google Earth platform was to have greater precision in the digitization process of the polygons.
This clarification is added in the corrected manuscript in lines 281-282 as follows:
Due to the time difference between the image used for integrating the GT and the date of the extraordinary rainfall (11 months); A validation was applied to ensure that the polygons identified in Google Earth existed in the 2013 ASTER image.
- Line 256-257, the GT inventory sampling, areas with surfaces greater than 450 m2 (two ASTER pixels). Here means, is 450m2 about two ASTER pixels? is it not 15 to 90 square meters per pixel of ASTER?
The spatial resolution of ASTER imagery is 15 meters, and therefore 1 pixel covers an area of 225 m2 (15m x 15m). In work, priority was given to identifying landslides greater than 2 pixels, that is, areas of 15m x 30m, which cover a surface of 450m2 (225m2 x 2 pixels).
- Line 258, Six hundred seventy-one should be expressed 671 for being consistent with the following expression.
The indicated correction is made.
- 12-13 are the close-up to the maps, which location should be given in the whole study area.
Figures 12 and 13 have been corrected following the instructions of the reviewer.
- The accuracy of the 5 models may related with the parameters used in the training of classifiers, which shown in Table 2. So how to select the most optimal parameters is a key problem. The selection of parameters in Table 2 should be described in detail. Alternatively, it is suggested that the impact of parameters be explained in the discussion
Indeed, the accuracy of classification models may vary according to different values of the parameters used when training the classifiers. We used the default parameters for each classifier for this study but decided to include Table 2 for reproducibility purposes.
As the reviewer mentioned, selecting the optimal parameters of a classifier is a key problem. Although some algorithms deal with this task (e.g., Grid search, Randomized search), we decided not to use any parameter selection algorithm in this study. We wish to highlight the classification methodology as the main contribution. It is evident that a careful selection of parameters may lead to better results. However, we are proposing a general methodology; thus, we wanted to report the results without “fine-tuning” the algorithms to a specific region, using them with their default parameters.
This clarification is added in the corrected manuscript in line 433 as follows:
All classification methods were run using their default parameters
Moreover, in lines 610-615 as follows:
As was previously mentioned, all classification methods were run using their default parameters. It is evident that a careful selection of parameters may lead the classification algorithms to better results regarding a specific region of interest. However, although there are some parameter selection methods (e.g., Grid search or Randomized search), we decided not to use any of them, as we wish to highlight this study as a generalized methodology for any other study area.

Reviewer 2 Report
This paper dealt with solved problems using traditional and available methods. There is no novelty in this paper.There are already very good ways to solve the problems mentioned in this paper in the existing literature. My recommendation is therefore to reject this manuscript.Author Response
Response to Reviewer 02
Manuscript ID: Remotesensing-1407105
Title: Evaluation of conditioning factors of slope instability and continuous change maps in the generation of landslide inventory maps using Machine Learning (ML) algorithms
Journal: Remote sensing
We thank the anonymous reviewers for their valuable comments and suggestions that helped improve our work.
REVIEWER 2
This paper dealt with solved problems using traditional and available methods. There is no novelty in this paper.There are already very good ways to solve the problems mentioned in this paper in the existing literature. My recommendation is therefore to reject this manuscript.
Response: We, the authors, broadly respect the reviewer's opinion who, from his particular point of view, says that “there is no novelty in this paper, and there are already very good ways to solve the problems mentioned.”
However, we believe that the perspective of the work and the methodology with which the solution of the proposed problem has been approached through evaluating the contribution of the conditioning factors addressed through the robustness of the algorithms of ML that allow studying the contribution as a whole of the variables involved, could be of interest to the community related to the study of Landslides, as evidenced by the comments of the other three manuscript reviewers, who believe that the work can be valuable and likely to be published; thus they also suggest changes to improve it before being published.

Reviewer 3 Report
Dear authors,
The manuscript you have submitted for publication represents a study on the use of machine learning methods for developing a landslide inventory.
I guess that the general topic is interesting as machine learning methods may provide a relevant contribution in the development of landslide inventories. In addition, in my opinion the results presented in the paper study are worth of being published after some minor revision in the “Introduction” and in “Materials and Methods” Sections (see following comments) and after an extensive editing of the English language and style.
MAIN COMMENTS
INTRODUCTION: When you introduce landslide inventories (page 2, lines 53-66) before moving on machine learning techniques you should first discuss more on the different other methodologies for compiling a landslide catalog: e.g., field surveys; information from reports, aerial photographs; information from newspaper and other online sources; There are several scientific papers dealing with this topic that you should cite in the Introduction.
MATERIALS AND METHODS: You state that slope angle, aspect and lithology are crucial factors for assessing slope instability. However, there are several other factors that should be evaluated (e.g., elevation, curvature, distance to rivers and roads) and the relevance of the factors depend on the types of landslides and the characteristics of the study area. How did you state that slope angle, aspect and lithology are the most relevant factors for your study area? Did you evaluate other factors in a preliminary phase?
English needs to be significantly improved, as several parts of the paper are difficult to follow and some terms are not appropriated.
SPECIFIC COMMENTS:
Page 1, line 31. Avoid acronyms the first time that you introduce a word in the text
Page 1, line 33: Change “Angular Slope” with “Slope angle” and “Terrain orientation” with “Aspect” here and throughout the paper
Page 1, line 34: Specify which is the zone in the southern part of Mexico
Page 1, line 34: What are you meaning? Landslide and non-landslide pixels?
Figure 3: the quality of the Figure should be increased
3.1 Data set: What is the resolution of the Lithology map? Is the same of the ASTER images and the DEM?
Page 7, lines 254-259: This part is not clear to me: how did you define landslide and non-landslide pixels? Which is the resolution of the pixels?
Author Response
Response to Reviewer 03
Manuscript ID: Remotesensing-1407105
Title: Evaluation of conditioning factors of slope instability and continuous change maps in the generation of landslide inventory maps using Machine Learning (ML) algorithms
Journal: Remote sensing
We thank the anonymous reviewers for their valuable comments and suggestions that helped improve the manuscript's quality and clarity. A detailed point-by-point response to the reviewers’ comments (in blue) is provided in the following paragraphs.
REVIEWER 3
- INTRODUCTION: When you introduce landslide inventories (page 2, lines 53-66) before moving on machine learning techniques you should first discuss more on the different other methodologies for compiling a landslide catalog: e.g., field surveys; information from reports, aerial photographs; information from newspaper and other online sources; There are several scientific papers dealing with this topic that you should cite in the Introduction.
Following the reviewer's suggestion, an analysis of other methodologies for compiling landslide inventory maps has been added. (Lines 73-81), as follows:
The approaches applied in integrating landslide inventory maps have evolved with technological advances that have allowed access to satellite images with better spatial, radiometric, and temporal resolutions. The inventories of landslides integrated from visual inspections and mapping of landslides on photographs or images, complemented through field studies, have been helpful in the integration of maps of hazards and susceptibility to landslides [14]. According to Harp et al. [15], in the early 1960s, the first earthquake-induced landslide inventory maps were made by using aerial photographs; However, access to satellite images has allowed the integration of landslide inventories applying semi-automatic methods [14, 16], detection of changes and image fusion [17].
And new references have been incorporated.
- MATERIALS AND METHODS: You state that slope angle, aspect and lithology are crucial factors for assessing slope instability. However, there are several other factors that should be evaluated (e.g., elevation, curvature, distance to rivers and roads) and the relevance of the factors depend on the types of landslides and the characteristics of the study area. How did you state that slope angle, aspect and lithology are the most relevant factors for your study area? Did you evaluate other factors in a preliminary phase?
The carried out process for the integration of this work is described below:
In previous works, other factors are analyzed, such as Deforestation, Distance and Density to roads, Lithology, Distance and Density to geological lineaments, Previous landslides, Slope angle, Aspect, Distance and Density to the flow network, Curvature, Profile curvature, Plan curvature, Topographic Wetness Index (TWI), Textural class, Potential erosion, Land Use/Cover, Accumulated precipitation, and Seismic activity.
Ramos-Bernal RN. (2018). Estudio de la susceptibilidad al deslizamiento de laderas en el Estado de Guerrero, México, aplicando Tecnologías de Información Geográfica. Ph.D. Thesis. Departamento de Tecnología Química y Ambiental. Rey Juan Carlos University. Campus Mostoles. Madrid, Spain. Link: https://burjcdigital.urjc.es/handle/10115/15869
In the same cited document and following the recommendations of other works, the individual or group effect that each of these factors has on the generation of landslides was evaluated (DeGraff, Romesburg 1980, Cooke, Doornkamp 1990, McDermid, Franklin 1995, Rojas Sequera 1996, Guzzetti et al. 1999, Dai, Lee 2002, Cevik, Topal 2003, Lee et al. 2004b, Ayalew, Yamagishi 2005, Gómez, Kavzoglu 2005, Yesilnacar, Topal 2005, Moreno et al. 2006, Ohlmacher 2007, Dahal et al. 2008, Nefeslioglu et al. 2008a, Yilmaz et al. 2012, Park et al. 2013, Chen et al. 2015, Marc et al. 2015, Rawat et al. 2015, Wang et al. 2017). Note: full references are included in the Ph.D. Thesis mentioned above.
This analysis was advised by a group of experts in geology and geomorphology from the Autonomous University of Guerrero (Mexico) and the Rey Juan Carlos University (Spain), with knowledge of the general characteristics and conditions of the study area. With the opinion on the interaction between each factor, a reduced group was defined: Lithology, Potential erosion index, Aspect, Density to geological lineaments, Slope angle, Plan curvature, Density to the flow network, Density to roads, and Potential effect of precipitation.
Finally, the individual contribution of each of the mentioned factors was evaluated through graphs of response curves and the Jackknife test. With the obtained results from this last analysis, it was possible to identify that the Slope angle, the Aspect, and the Lithology are crucial factors to evaluate the terrain instability for the particular study area addressed in this work (Ramos-Bernal, 2018).
In the manuscript, the discussion of the factors is addressed in the Introduction, in the topic 1.1 Conditioning factors of terrain stability.
To clarify the definition of the factors used in this work, the corresponding paragraph (Lines 113-116) has been supplemented as follows:
Some of the most common factors among the analyzed works are slope angle, aspect, and lithology, which are included in this study mainly based on an analysis of the individual contribution of each of the mentioned factors, carried out through graphs of response curves. and the Jackknife test, developed by Ramos-Bernal (2018)
- English needs to be significantly improved, as several parts of the paper are difficult to follow and some terms are not appropriated.
The original manuscript has been sent to the professional English editing service and adjusted of the style according to the publication needs of the journal.
The certificate of the professional edition of English is attached.
- Page 1, line 31. Avoid acronyms the first time that you introduce a word in the text.
This observation has been addressed in the corrected manuscript (Lines 35-37) as follows:
This work presents the performance of five machine learning methods—K-Nearest Neighbor (KNN), Stochastic Gradient Descendent (SGD), Support Vector Machine Radial Basis Function (SVM RBF Kernel), Support Vector Machine (SVM Linear Kernel), and AdaBoost—
- Page 1, line 33: Change “Angular Slope” with “Slope angle” and “Terrain orientation” with “Aspect” here and throughout the paper.
Following the instructions of the reviewer, the variable names have been changed throughout the text and figures 3 and 4 in the corrected manuscript.
- Page 1, line 34: Specify which is the zone in the southern part of Mexico
Following the reviewer's indication, the information is supplemented in the corrected manuscript (Line 38) as follows:
… in a zone of the state of Guerrero in southern Mexico,
Furthermore, the zone is described in detail and is shown in Figure 1 in topic 2. Study area.
- Page 1, line 34: What are you meaning? Landslide and non-landslide pixels?
Indeed, the text "... samples of 671 slide / non-slide polygons." refers to digitized samples of slipped and non-slipped polygons; which are subsequently treated in raster format as pixels.
We consider that the text may be clearer "samples of 671 slidden / non-slidden polygons" so it has been modified in the corrected manuscript (lines 40-41).
- Figure 3: the quality of the Figure should be increased
Figure 3 was modified in the corrected manuscript
- 3.1 Data set: What is the resolution of the Lithology map? Is the same of the ASTER images and the DEM?
All the maps of the variables used (Figure 4) have the exact spatial resolution as the ASTER and DEM images (15 meters).
- Page 7, lines 254-259: This part is not clear to me: how did you define landslide and non-landslide pixels? Which is the resolution of the pixels?
Landslide pixels refer to the pixels of the images or maps contained within the polygons identified as real landslides.
On the other hand, the non-landslide pixels refer to the images or maps contained within the polygons identified as stable or non-landslide areas in the analyzed period.
In all cases, the spatial resolution, that is, the size that a pixel represents on the ground, is 15 meters.

Reviewer 4 Report
remotesensing-1383467-peer-review-v1
The manuscript “Evaluation of conditioning factors of slope instability and continuous change maps in the generation of landslide inventory maps using Machine Learning (ML) algorithms” addresses an interesting and up-to-date subject, which adheres to Remote Sensing journal policies.
Although the manuscript does not have high novelty, it presents a good RS application and contains interesting results. In addition, the work is well conceived and written, with good figures and an interesting and worthwhile case study.
In my opinion, the manuscript must have some improvements before publication:
- Additional information in M&M regarding the software used
- Discussion chapter should be improved and citations should be added
- The paragraphs from conclusions that contain citations should be moved to Discussion
Author Response
Response to Reviewer 04
Manuscript ID: Remotesensing-1407105
Title: Evaluation of conditioning factors of slope instability and continuous change maps in the generation of landslide inventory maps using Machine Learning (ML) algorithms
Journal: Remote sensing
We thank the anonymous reviewers for their valuable comments and suggestions that helped improve the manuscript’s quality and clarity. A detailed point-by-point response to the reviewer’s comments (in blue) is provided in the following paragraphs.
REVIEWER 4
The manuscript “Evaluation of conditioning factors of slope instability and continuous change maps in the generation of landslide inventory maps using Machine Learning (ML) algorithms” addresses an interesting and up-to-date subject, which adheres to Remote Sensing journal policies.
Although the manuscript does not have high novelty, it presents a good RS application and contains interesting results. In addition, the work is well conceived and written, with good figures and an interesting and worthwhile case study.
We, the authors, appreciate and welcome the reviewer’s comments.
- Additional information in M&M regarding the software used?
In general, on the subject of materials and methods of the original manuscript, the software used is mentioned (Lines 261 and 262). However, in the corrected manuscript; the information corresponding to the software has been added with more detail in lines 224-249 as follows:
- The first step consists of acquiring primary data: ASTER images, Digital Elevation Model (DEM), geological-mining charts, aerial photographs, and field data. Derived maps were generated from Principal Component 1 (PC1) in Principal Component Analysis (PCA), Normalized Difference Vegetation Index (NDVI), cloud masks, Angular Slope (AS), Terrain Orientation (TO), Lithology (L), and Ground Truth (GT) samples. Dinamica EGO 5 was used to generate the NDVI images; the PCA images were obtained using ArcMap 10.3;
- Automatic change detection. At this stage, two change detection methods were applied, Linear Regression (LR) and Image Differencing (Diff), to the maps derived from the ASTER images. Three maps of continuous change were generated: LR applied to PC1 (LR-PC1), LR applied to NDVI (LR-NDVI), and Diff applied to PC1 (Diff-PC1). This process was performed with Dinamica EGO 5;
- Supervised classification. In this stage, the K-Nearest-Neighbor (KNN), Stochastic Gradient Descent (SGD), Support Vector Machine (SVM), and AdaBoost classifiers were applied using the previously obtained continuous change maps (LR-PC1, LR-NDVI, and Diff-PC1) and the factors of slope stability (AS, TO, and L) considered. The process was repeated and complemented by incorporating the factors into the classification, one by one, and combining them. All classification algorithms, so as the accuracy evaluation metrics were run using the scikit-learn version 0.22.1 for Python 3.6.5 programming language;
- Accuracy assessment. In this stage, the inventory maps obtained by supervised classification were evaluated by confusion matrices, omission and commission errors, and metrics such as the Kappa concordance coefficient (k), Accuracy (ACC), Precision, Recall, and F1 score. The Python’s Georasters library version 0.5.20 was used to read the derived satellite images, including its metadata, and generate the maps after the classification stage.
Thus, the original description of the software (Lines 261 and 262) has been eliminated with the aim of not repeating it.
- Discussion chapter should be improved and citations should be added.
A paragraph was added to the topic of discussion, including citations (lines 616-622) as follows:
Some other works apply well-known deep learning architectures [19, 20, 84–86]. Chen et al. [19] and HacıefendioÄŸlu et al. [20] in their results report an average recall of 0.86 and 0.91, respectively; Lu et al. [84], Defang et al. [85] and Qin et al. [86]; report an accuracy of 0.98; 0.98 and 0.88 respectively. As can be seen, their results are very similar to those achieved in this work (Table 4). However, their algorithms require higher computational costs than those presented in this study.
- The paragraphs from conclusions that contain citations should be moved to Discussion.
The reviewer's suggestion in the corrected manuscript has been heeded

Round 2
Reviewer 2 Report
I keep my previous opinion. I don't think this paper can be published.